# A method to estimate the contribution of rare coding variants to complex trait heritability

Nazia Pathan [1,2], Wei Q. Deng [3,4], Matteo Di Scipio[1,5], Mohammad Khan [1,5], Shihong Mao[1], Robert W. Morton [1,2], Ricky Lali[1,6], Marie Pigeyre [1,5], Michael R. Chong[1,2,7] & Guillaume Paré [1,2,6,7] ✉

It has been postulated that rare coding variants (RVs; MAF < 0.01) contribute to the "missing" heritability of complex traits. We developed a framework, the Rare variant heritability (RARity) estimator, to assess RV heritability ($h^2_{RV}$) without assuming a particular genetic architecture. We applied RARity to 31 complex traits in the UK Biobank ($n = 167,348$) and showed that gene-level RV aggregation suffers from 79% (95% CI: 68-93%) loss of $h^2_{RV}$. Using unaggregated variants, 27 traits had $h^2_{RV} > 5\%$, with height having the highest $h^2_{RV}$ at 21.9% (95% CI: 19.0-24.8%). The total heritability, including common and rare variants, recovered pedigree-based estimates for 11 traits. RARity can estimate gene-level $h^2_{RV}$, enabling the assessment of gene-level characteristics and revealing 11, previously unreported, gene-phenotype relationships. Finally, we demonstrated that in silico pathogenicity prediction (variant-level) and gene-level annotations do not generally enrich for RVs that over-contribute to complex trait variance, and thus, innovative methods are needed to predict RV functionality.

Rare protein coding variants, herein defined as those having minor allele frequency (MAF) < 1% and referred to as RVs, represent an important and understudied component of non-Mendelian complex trait genetics[1]. Despite efforts to functionally characterize RVs, the biological impact of roughly 400 rare, putatively disruptive mutations carried by each individual remains largely unknown[2]. Classification of RVs is challenging, and current algorithms do not always correctly predict their pathogenic characteristics[3,4]. Indeed, existing tools to classify RVs have typically been trained on conditions of Mendelian inheritance[5–7], while most human phenotypes are complex and non-Mendelian in nature. This gives rise to an unmet need to assess existing classifications in the context of complex traits.

Genome-wide association studies (GWAS) have been fruitful for characterizing common variants with regards to complex traits; however, a similar approach lacks the statistical power to study rare variants, unless sample sizes or effect sizes are very large[7]. Consequently, to improve statistical power, RV association testing often relies on gene-level variant aggregation methods to perform gene-burden tests, or variance component tests such as Sequence Kernel Association Test (SKAT)[8] and its variations (e.g., SKAT-O or "Optimal SKAT"). A limitation of gene-burden tests is the assumption that all RVs influencing a trait are homogeneous in terms of direction and magnitude of effects[7]. SKAT, on the other hand, aggregates the associations between variants and the phenotype through a kernel matrix[8], is a powerful tool for

[1]Population Health Research Institute, David Braley Cardiac, Vascular and Stroke Research Institute, Hamilton Health Sciences and McMaster University, Hamilton, Canada. [2]Department of Pathology and Molecular Medicine, McMaster University, Michael G. DeGroote School of Medicine, Hamilton, Canada. [3]Peter Boris Centre for Addictions Research, St. Joseph's Healthcare Hamilton, Hamilton, Canada. [4]Department of Psychiatry and Behavioural Neurosciences, McMaster University, Hamilton, Canada. [5]Department of Medicine, Faculty of Health Sciences, McMaster University, Hamilton, Canada. [6]Department of Health Research Methods, Evidence, and Impact, McMaster University, Hamilton, Canada. [7]Thrombosis and Atherosclerosis Research Institute, David Braley Cardiac, Vascular and Stroke Research Institute, Hamilton, Canada. ✉e-mail: pareg@mcmaster.ca

association testing in the presence of variants acting in opposing directions, but can be less powerful than burden tests when most variants are causal, and effects are in the same direction. SKAT-O combines both burden and SKAT to overcome the limitations of SKAT but can be slightly less powerful than burden or variance-component tests if their assumptions are held. While SKAT and SKAT-O are solely designed for association testing, variant aggregation in gene-burden testing may be used in any scenario requiring a gene to be treated as a single unit. However, the impact of RV aggregation on phenotypic variance explained has never been empirically evaluated.

RVs are postulated to contribute significantly to the "missing heritability" of complex traits, i.e. rare coding variants, when combined with common variants (CVs) may help recover the difference between current SNP-based heritability and heritability estimates from the traditional pedigree-based studies[9,10], yet this hypothesis has only been assessed for a handful of traits[11–17]. The RV contribution to narrow-sense heritability, $h^2_{RV}$, defined as the proportion of phenotypic variance attributable to their additive genetic effects, has been estimated in several recent studies[11–17]. However, these studies are limited by either the use of genotyping array data, small sample size, or models that make specific assumptions about the underlying genetic architecture. RV heritability estimates utilizing genotype data[12,13,15,16] are limited by the selection of rare genetic variation captured on the array, and algorithms to impute missing variants resulted in low accuracy for truly rare alleles[16,18]. As next-generation sequencing, particularly, whole exome sequencing (WES), are becoming a common place to accurately detect RVs[11,14,17], methods to evaluate RV heritability that do not rely on assumptions about the genomic architecture are needed.

We propose an approach to estimate RV narrow-sense heritability ($h^2_{RV}$), the Rare-variant heritability (RARity) estimator. A common strategy to estimate heritability is by assuming a random genetic effect model, whereby each (unknown) causal variant contributes to the total phenotypic variance according to a statistical distribution that might depend on its MAF, linkage disequilibrium (LD) with nearby variants, and functional properties, among others. These models aim to evaluate the overall contribution from a large number of variants, without identifying variant-specific effects[19]. RARity, in contrast, is based on multiple linear regression to estimate heritability, where the genetic effects are estimated conditional on the observed genotypes, and thus does not make any assumptions about the distribution of individual genetic effects nor the joint distribution of genotypes. The RARity framework simultaneously evaluates a large number of smaller regions of the chromosomes, referred to as blocks. This step is necessary to avoid estimation of high-dimensional linear models as the large number of rare variants would otherwise make matrix calculations intractable even with modern computational capabilities. An important and critical feature of RARity is the pruning of variants to minimize inflation in heritability due to LD spillage between blocks (see Methods).

The availability of large WES data from the UK Biobank (UKB) provides a unique opportunity to study the overall and gene-level contribution of RVs to complex trait heritability. We hypothesize that the study of exome-wide and gene-level RV heritability will provide insights into the functional characteristics of RVs. Specifically, by estimating the contribution of rare coding variants to narrow-sense heritability in unrelated individuals, our objectives were to (1) characterize the variant-level characteristics (allele frequency, disruptiveness according to a variety of algorithms, and clinical pathogenicity) that best predict phenotypic variations, (2) characterize the gene-level properties (membership to gene sets and biological pathways, evolutionary constraint, gene length) that best predict biological effects and, (3) identify genes associated with the traits.

Using WES data from UK Biobank ($n = 167,348$), we report $h^2_{RV}$ for 31 complex continuous traits, including 26 biomarkers and 5 anthropometric traits, and we demonstrate the utility of RARity estimator to understand whether existing in silico pathogenicity prediction (variant-level) and gene-level annotations could be enriched for RVs that disproportionately contribute to the complex trait. This study has major implications for our understanding of the genetic architecture of complex traits in the context of RV functions, which we expect would ultimately facilitate the discovery of new disease pathogenesis.

## Results

### Overview and testing of the RARity method

The RARity method entails parallel computing of the adjusted $R^2$ based on an ordinary least square (OLS) multiple linear regression as an unbiased estimator of block-wise heritability for each consecutive genetic block. Adjusted $R^2$ estimates are then summed over all blocks as the overall heritability estimate. Overview of RARity is shown in Fig. 1 and technical details provided in Methods and Supplementary Fig. 1. The current sample size provided at least 80% power to detect 4% $h^2_{RV}$, at an empirical type-I error rate of 0.05 (Supplementary Fig. 2). Extensive simulations were performed using real genotype data to identify an approach for estimating heritability that is robust under realistic scenarios. Through this endeavour, we discovered that $h^2_{RV}$ could be affected by the presence of long-range LD (LRLD), occurring at a much greater distance than what is observed for CVs[20] (Supplementary Fig. 3). LRLD complications were controlled by using a stringent LD threshold over a large, empirically derived window size ($r^2 > 0.1$, window size = 50 Mb, step size = 500 bases). This stringent pruning method was enforced in all subsequent analyses involving RVs to ensure a well-calibrated estimate of $h^2_{RV}$. The simulation studies on the effects of varying MAFs, heritability, proportion of causal genes or variants indicated that RARity is largely unbiased but tends to underestimate when the number of causal variants or genes is low (<1%; Supplementary Fig. 3).

### Comparison of gene-burden, gene-wise, and exome-wide heritability

To estimate the amount of information lost when aggregating rare variants within each gene, we compared $h^2_{RV}$ estimates using the gene-burden approach to exome-wide estimations for 31 complex traits using RARity. We created blocks of genotype data in the following manners: (1) gene-burden blocks, derived by summing the number of rare alleles within each gene for an individual, which produced a single block containing all gene-burden scores as predictors; (2) gene-wise blocks, consisted of un-aggregated RVs partitioned by gene, such that each block contained all the variants within a single gene; (3) exome-wide blocks were created by partitioning RVs in each chromosome into blocks of ~5000 adjacent RVs. RV heritability estimates were then derived from each type of block construct using RARity. Design and applications of RARity, using these constructs is provided in Fig. 1, and a detailed computational pipeline is illustrated in Supplementary Fig. 1.

Our results show that the overall estimated RV heritability from gene-burden blocks ($h^2_{RV-burden}$) is on average 79.3% (95% CI: 76.5%–82.0%) less than heritability based on either gene-wise ($h^2_{RV-gene-tot}$), or exome-wide blocks ($h^2_{RV}$; Fig. 2, Supplementary Table 3). Because gene-level estimates ($h^2_{RV-gene}$) are useful for secondary analyses, we further tested whether $h^2_{RV-gene-tot}$ could in fact substitute for $h^2_{RV}$ from exome-wide blocks. A potential caveat of using gene-wise heritability to determine total heritability is the possibility of LD between variants in two or more genes inflating the total heritability, as would be the case when genes overlap. We observed consistent results between $h^2_{RV-gene-tot}$ and $h^2_{RV}$ (Fig. 2b and Supplementary Table 3), with $h^2_{RV-gene-tot}$ being 1.5% (95% CI: 0.0%–4.0%) higher than $h^2_{RV}$ on average, indicating only a slight inflation that could be due to chance. However, the advantage of gene-level blocks is the smaller number of variants per block, which makes it computationally less intensive and 3× faster to compute than $h^2_{RV}$.

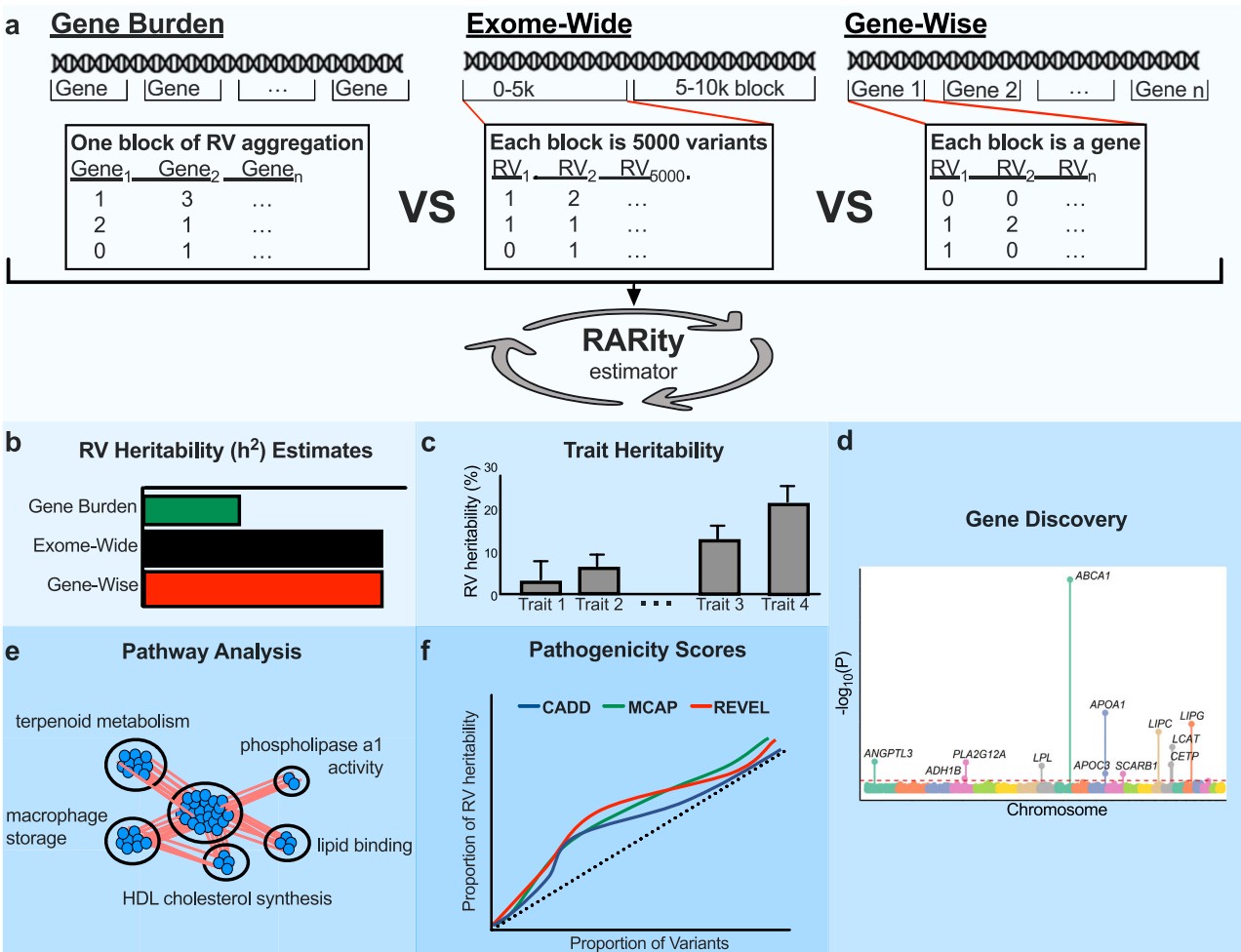

**Fig. 1 | A schematic diagram illustrating the design and applications of Rare variant heritability (RARity) estimator.** Following the initial quality control steps (Methods), the protein-altering and LoF RVs in each chromosome were LD pruned and partitioned into consecutive regions, referred to as genotype blocks. **a** Diagrammatic representation of the three genotype block constructs implemented by RARity. Applications of RARity included: **b** Comparison of the block constructs to empirically test the loss of information when aggregating rare variant heritability estimates of complex traits using exome-wide blocks. **d** Gene-based RV heritability to discover gene-phenotype associations. **e** Network and pathway analyses of the genes with significant RV heritability to provide insights into their biological relevance. **f** Enrichment of RV heritability by pathogenicity scores. Three pathogenicity scores, CADD, M-CAP and REVEL were applied to retain progressively more deleterious variants and tested on all 31 traits for enrichment in RV heritability. This figure was generated using GraphPad Prism version 6.04 for Windows[66].

## RV contribution to heritability estimates for 31 complex traits

Here we utilized two genetic datasets from the UK Biobank: RVs were derived from the WES data (UKB data field: 23155) and CVs were extracted from the imputed genotype data (data field: 22418; v3 release). The quality control steps for each dataset are provided in Methods. Applying RARity using the exome-blocks (MAF < 0.01), we estimated RV heritability for 31 complex traits, with 29 traits showing meaningful contribution at a nominal significant threshold of 0.05, 27 traits showing $h^2_{RV}$ > 5% and height having the highest $h^2_{RV}$ at 21.9% (95% CI: 19.0%–24.8%) (Table 1; Fig. 3, and Supplementary Table 3). The lower overall heritability estimate for glucose, 1.8% (95% CI: −1.0%–4.7%), observed in our study was most likely due to sample collection in non-fasting states, as opposed to the fasting glucose level used for heritability estimates in the pedigree studies[21]. Sex stratification, performed on all 31 traits, showed some heterogeneity in $h^2_{RV}$ between the sexes (Supplementary Fig. 4), but the apparent differences were statistically nonsignificant (p-value > 0.05).

Since RARity poses no upper restriction on the MAFs, we additionally assessed the contribution CV ($h^2_{CV}$) and the combined CV and RV ($h^2_{tot}$) to these 31 traits by concatenating common and rare variants (Methods). We observed that $h^2_{CV}$ estimated using RARity was

consistent with BOLT[22], but higher than LDSC[23] (Supplementary Fig. 5). Although common variants contributed more to overall heritability, as compared to RVs (Table 1), estimated $h^2_{RV}$ generally increased proportionally to $h^2_{CV}$ (Supplementary Fig. 6), except for height, alkaline phosphatase and Lp(a). Lp(a) particularly stands out with a much higher $h^2_{CV}$ in relation to $h^2_{RV}$. The low concordance between CV and RV heritability in Lp(a) may be due to the unique genetic architecture of this trait, with most genetic variance attributed to the *LPA* locus itself and the highly polymorphic kringle IV type 2 copy number variation having an outsized impact on concentration[24].

The estimated $h^2_{tot}$ for 11 of the 31 traits were consistent with previously reported heritability from pedigree or twin-based studies (Supplementary Data 1). The difference between $h^2_{RV}$ and $h^2_{RV}$-adjusted-for-$h^2_{CV}$ were heterogenous (Table 1), indicating that the degree of tagging or LD between the rare and common variants can vary by trait. For example, $h^2_{tot}$ (from CV and RV) was 87.8% (95% CI: 84.5–91.1%) for height and 22.5 (95% CI: 19.3–25.7%) for albumin, with height showing a greater degree of LD between RVs and CVs, as compared to albumin, determined by a 5.3% reduction in $h^2_{RV}$ for height, but only 0.6% reduction in $h^2_{RV}$ for albumin following adjustment for $h^2_{CV}$ (Table 1).

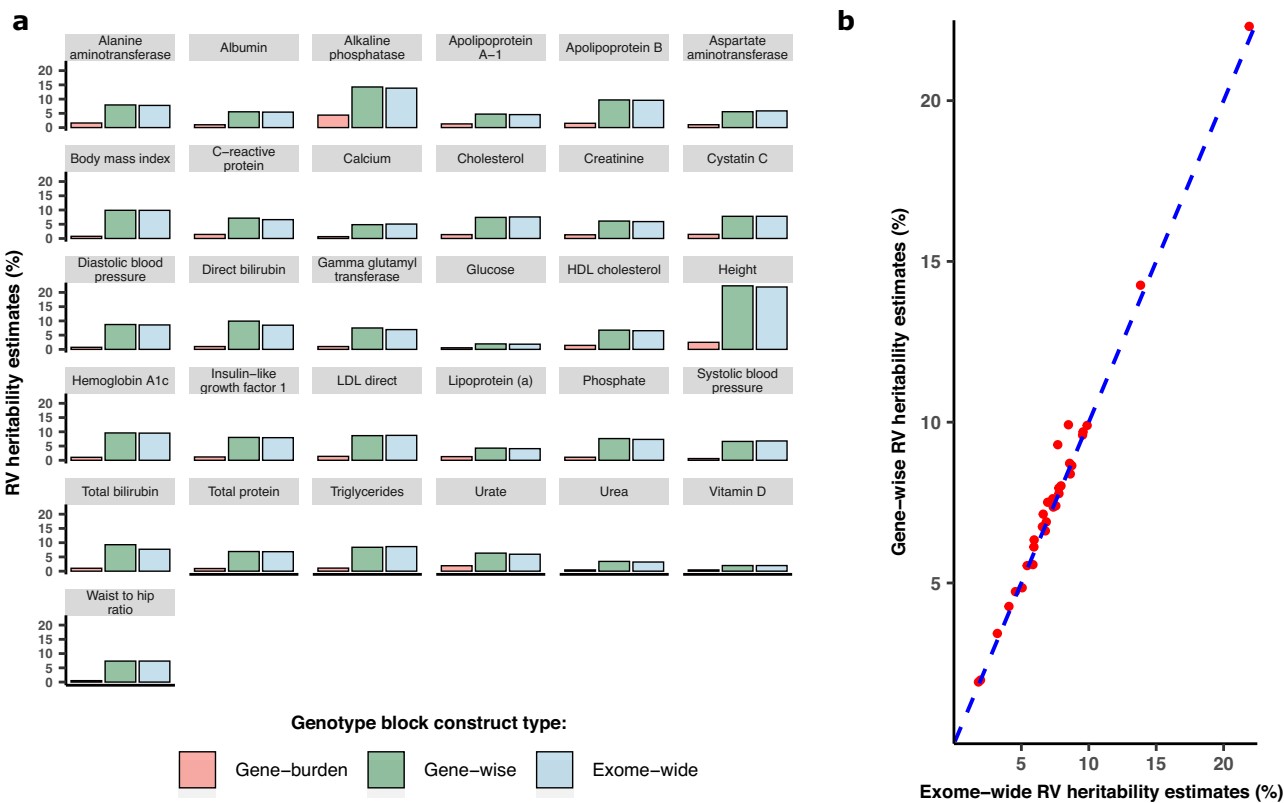

**Fig. 2 | Effect of aggregating variants on RV heritability estimates. a** Comparison of RV heritability estimates derived from aggregation of variants in gene-burden blocks, exome-wide blocks (blocks of 5000 unaggregated variants) and gene-wise blocks with un-aggregated variants. The *y*-axis corresponds to the rare coding variant contribution to percentage of heritability estimates for each trait. **b** Correlation between heritability estimates from gene-wise blocks represented on the *y*-axis versus exome-wide blocks, represented on the *x*-axis.

## RV heritability to characterize pathogenicity scores

Various in silico tools have been developed to predict the effects of rare coding variants on risk of Mendelian disorders. Whether such pathogenicity tools are useful for prioritizing variants contributing to complex trait variance in the general population remains uncertain. As such, we tested the association between commonly used pathogenicity scores, namely Combined Annotation-Dependent Depletion (CADD), Mendelian Clinically Applicable Pathogenicity (MCAP), and Rare Exome Variant Ensemble Learner (REVEL), and the fraction of trait RV heritability explained, hypothesizing that variants predicted to be more deleterious account for more trait variance explained and vice versa.

For most traits, the fraction of heritability explained by an increasing proportion of RVs (added from the highest to the lowest level of predicted pathogenicity), was largely uniform and independent of pathogenicity score (sub-plot Fig. 4, and Supplementary Fig. 7), indicating little enrichment of trait-associated RVs by pathogenicity score. Across traits (Supplementary Fig. 7), the average RV heritability explained by the top 25% most deleterious variants was slightly higher than expected at 36% (CADD 34%; MCAP 36.7%; REVEL 37.2%). However, the top 50% of most deleterious variants accounted for 42.9% of heritability explained, which is lower than the fraction of heritability explained by the bottom 50% set of variants (54%). The magnitude of enrichment by pathogenicity score also varied by trait, for example, the top 25% most deleterious variants by M-CAP score explained as little as 32.3% RV heritability for height but as much as 48.4% RV heritability for ApoA-I. Further, the allele frequency of variants seemed to have very little impact on the magnitude of enrichment, as we observed similar results across the three MAF categories (<0.001, <0.005, <0.01; Fig. 4, Supplementary Fig. 7). The specified MAF cutoffs were used instead of MAF-bins as the $h^2_{RV\text{-}gene\text{-}tot}$ in these bins

(0.01 > MAF ≥ 0.005, 0.005 > MAF ≥ 0.001, and 0.01 > MAF ≥ 0.001) were <5% (Supplementary Table 4) and consequently would have produced unreliable results when further stratified by pathogenicity scores.

## Characterizing genes based on heritability estimates

To investigate gene-level characteristics of RVs, we used RARity to determine $h^2_{RV\text{-}gene}$ for all genes with qualifying variants and derived corresponding *p*-values for each gene using an *F*-test. After Bonferroni correction (*p*-value < 2.75 × 10⁻⁶), 152 of the 18,214 genes had significant $h^2_{RV\text{-}gene}$ for one or more traits (herein referred to as significant $h^2_{RV\text{-}gene}$), representing 218 distinct gene-biomarker relationships. A list of these genes with the corresponding $h^2_{RV\text{-}gene}$ and *p*-values are presented in Supplementary Data 2. We identified many genes that recapitulated previously reported associations and discovered 11 previously unidentified gene-traits relationships. Some examples of well-established relationships with significant $h^2_{RV\text{-}gene}$ include *PCSK9*, which is known to regulate Apolipoprotein B (ApoB) and Low density cholesterol (LDL); *MC4R* gene affecting body mass index (BMI), and the association of *LPL* with Apolipoprotein A-I (ApoA-I) levels. Some of the previously unidentified relationships include *PPARA* with ApoB, *TFAM* with alkaline phosphatase, *TMEM43* with hemoglobin A1c (HbA1c), and *NR1I2* with total bilirubin. A list of the significant heritability genes, highlighting the gene-biomarker relationships is provided in Table 2. In addition, Manhattan plots (Fig. 5) are included for 6 randomly selected traits: Alkaline phosphatase, HbA1c, Low density lipoprotein direct (LDL), Insulin like growth-factor 1 (IGF-1), and ApoA-I.

We further investigated the role of the 152 genes contributing significantly to trait heritability (*p*-value < 2.75 × 10⁻⁶) in diseases and

**Table 1 | Heritability estimates derived from rare coding and common variants**

| Traits | $h^2_{RV}$ (%) (95% CI) | $h^2_{CV}$ (%) (95% CI) | $h^2_{tot}$ (%) (95% CI) | $h^2_{RV}$ adjusted for $h^2_{CV}$ (%) (95% CI) | Difference between $h^2_{RV}$ and $h^2_{RV}$ adjusted for $h^2_{CV}$ (%) |
|---|---|---|---|---|---|
| Albumin | 5.4 (2.6,8.3) | 17.7 (15.5,20.0) | 22.5 (19.3,25.7) | 4.8 (1.7,7.8) | 0.6 |
| Alkaline phosphatase | 13.8 (11.0,16.7) | 36.1 (33.9,38.4) | 46.8 (43.6,50.1) | 10.7 (7.6,13.8) | 3.1 |
| Alanine aminotransferase | 7.8 (4.9,10.6) | 17.1 (14.8,19.3) | 23.0 (19.7,26.2) | 5.9 (2.8,9.0) | 1.9 |
| Apolipoprotein A | 4.6 (1.7,7.4) | 27.1 (24.8,29.4) | 29.9 (26.6,33.1) | 2.8 (−0.3,5.8) | 1.8 |
| Apolipoprotein B | 9.6 (6.7,12.5) | 38.9 (36.6,41.2) | 44.0 (40.8,47.3) | 5.1 (2.0,8.2) | 4.5 |
| Aspartate aminotransferase | 5.9 (3.0,8.7) | 19.9 (17.6,22.1) | 24.0 (20.8,27.3) | 4.2 (1.1,7.2) | 1.7 |
| Body mass index | 9.9 (7.0,12.8) | 31.8 (29.6,34.1) | 39.5 (36.3,42.8) | 7.7 (4.6,10.7) | 2.2 |
| Calcium | 5.1 (2.2,7.9) | 16.1 (13.9,18.4) | 20.7 (17.5,24.0) | 4.6 (1.6,7.7) | 0.4 |
| Cholesterol | 7.6 (4.7,10.4) | 28.9 (26.6,31.2) | 32.2 (29.0,35.4) | 3.3 (0.2,6.3) | 4.3 |
| Creatinine | 5.9 (3.1,8.8) | 28.6 (26.3,30.9) | 32.6 (29.3,35.8) | 3.9 (0.9,7.0) | 2.0 |
| C-reactive protein | 6.6 (3.8,9.5) | 27.3 (25.0,29.6) | 33.7 (30.5,37.0) | 6.4 (3.4,9.5) | 0.2 |
| Cystatin C | 7.8 (4.9,10.7) | 33.5 (31.3,35.8) | 38.9 (35.7,42.2) | 5.4 (2.3,8.5) | 2.4 |
| Diastolic Blood Pressure | 8.6 (5.7,11.4) | 26.1 (23.9,28.4) | 34.6 (31.4,37.8) | 8.5 (5.4,11.5) | 0.1 |
| Direct bilirubin | 8.5 (5.6,11.4) | 35.0 (32.7,37.3) | 39.4 (36.1,42.6) | 4.4 (1.3,7.4) | 4.1 |
| Gamma glutamyl transferase | 7.0 (4.1,9.8) | 29.4 (27.1,31.7) | 35.0 (31.7,38.2) | 5.6 (2.5,8.6) | 1.4 |
| Glucose | 1.8 (−1.0,4.7) | 9.8 (7.6,12.1) | 12.5 (9.3,15.7) | 2.7 (−0.4,5.7) | -0.8 |
| Height | 21.9 (19.0,24.8) | 71.3 (69.0,73.6) | 87.8 (84.5,91.1) | 16.5 (13.5,19.6) | 5.4 |
| Hemoglobin A1c | 9.5 (6.7,12.4) | 31.4 (29.1,33.6) | 38.3 (35.1,41.6) | 7.0 (3.9,10.0) | 2.6 |
| HDL cholesterol | 6.6 (3.7,9.4) | 31.5 (29.2,33.7) | 36.2 (32.9,39.4) | 4.7 (1.6,7.8) | 1.9 |
| IGF-1 | 7.9 (5.1,10.8) | 28.1 (25.9,30.4) | 33.8 (30.5,37.0) | 5.6 (2.6,8.7) | 2.3 |
| LDL direct | 8.7 (5.9,11.6) | 32.4 (30.2,34.7) | 36.7 (33.5,39.9) | 4.2 (1.2,7.3) | 4.5 |
| Lipoprotein (a) | 4.1 (1.2,6.9) | 61.1 (58.9,63.4) | 61.6 (58.4,64.8) | 0.5 (−2.6,3.5) | 3.6 |
| Phosphate | 7.3 (4.5,10.2) | 17.1 (14.8,19.4) | 23.3 (20.1,26.6) | 6.2 (3.2,9.3) | 1.1 |
| Systolic Blood Pressure | 6.8 (3.9,9.6) | 25.4 (23.1,27.7) | 33.4 (30.2,36.7) | 8.0 (5.0,11.1) | −1.3 |
| Total bilirubin | 7.7 (4.8,10.6) | 40.3 (38.0,42.6) | 44.3 (41.1,47.6) | 4.1 (1.0,7.1) | 3.6 |
| Total protein | 6.9 (4.0,9.7) | 23.0 (20.7,25.2) | 29.4 (26.1,32.6) | 6.4 (3.3,9.5) | 0.4 |
| Triglycerides | 8.6 (5.8,11.5) | 26.3 (24.0,28.5) | 31.9 (28.6,35.1) | 5.6 (2.5,8.7) | 3.0 |
| Urea | 3.2 (0.4,6.1) | 17.1 (14.8,19.4) | 20.0 (16.8,23.2) | 2.9 (−0.2,6.0) | 0.3 |
| Urate | 6.0 (3.1,8.8) | 28.0 (25.7,30.3) | 32.5 (29.3,35.8) | 4.5 (1.5,7.6) | 1.4 |
| Vitamin D | 2.0 (-0.9,4.8) | 13.3 (11.0,15.5) | 15.0 (11.7,18.2) | 1.7 (−1.4,4.7) | 0.3 |
| Waist to hip ratio | 7.4 (4.5,10.2) | 21.0 (18.7,23.2) | 28.2 (25.0,31.5) | 7.2 (4.2,10.3) | 0.1 |

*CI* confidence interval, $h^2_{CV}$ heritability estimates due to common variants, $h^2_{RV}$ heritability estimates due to rare coding variants, $h^2_{tot}$ heritability estimates due to combined rare coding and common variants.

biological pathways. DisGenet[25] revealed 115 of the target genes influencing 2137 disease pathways (Supplementary Data 3). A heatmap of the diseases associated with the significant genes for ApoB is presented in Fig. 6a. Interestingly, searching through the Drug Gene Interaction database (DGIdb)[26] revealed that 93 of the 152 of the target genes (i.e., 61%) belong in the "druggable genome" category, 16 of which are "clinically actionable", including *APOB, FGFR3* and *LDLR* (Supplementary Data 4). The target genes also appear to be significantly overrepresented in many biologically relevant pathways (multiple-testing correction via the g:Profiler g:SCS algorithm, adjusted *p*-value < 0.05). For example, the genes contributing significantly towards $h^2_{RV}$ of ApoB results in enrichment of 156 pathways, all of which are highly interconnected in an elaborate network and includes well-known pathways such as the LDL receptor binding pathways and pathways related to atherosclerosis (Fig. 6b, Supplementary Data 5).

We explored whether $h^2_{RV-gene}$ is associated with gene-length and evolutionary constraint, where gene length was derived from the RefSeq transcripts with the greatest length and gene-level evolutionary constraint was determined using gnomAD pLoF Metrics[27]. For most traits, neither gene-length (Supplementary Fig. 8, Supplementary Table 5) nor evolutionary constraint (Supplementary Fig. 9, Supplementary Table 6) was significantly associated with $h^2_{RV-gene}$. Height was the only trait where $h^2_{RV-gene}$ was significantly associated with gene-length (0.24% variance explained, *p*-value = 3.3 × 10^{-11}) and evolutionary constraint (0.27% variance explained, *p*-value = 1.5 × 10^{-11}). Evolutionary constraint was suggestively associated with the $h^2_{RV-gene}$ of BMI (*p*-value = 3.7 × 10^{-4}) and waist-to-hip ratio (*p*-value = 9.4 × 10^{-3}), but no other significant association was observed. The highly conserved gene cluster regions, with short repeats, such as the hox, histone, protocadherin and hemoglobin gene clusters did not contribute significantly to $h^2_{RV}$ (Supplementary Table 7). The overall heritability estimates, $h^2_{RV-gene-tot}$, were consistent between genes that are transcribed from either the positive or the negative strand (Supplementary Fig. 10), which is not surprising, considering that there are nearly equal number of genes on either strand.

## Discussion

We established a method, the RARity estimator, to accurately estimate the contribution of RVs to the heritability of complex quantitative traits, and to characterize the gene-level and variant level characteristics of RVs, as demonstrated with results from 31 continuous traits form the UK Biobank. RARity is a versatile method, as it does not make any prior assumptions about the genetic architecture of the selected variants, making it applicable to both common and rare variants. Calibration of RV contributions was empirically confirmed using

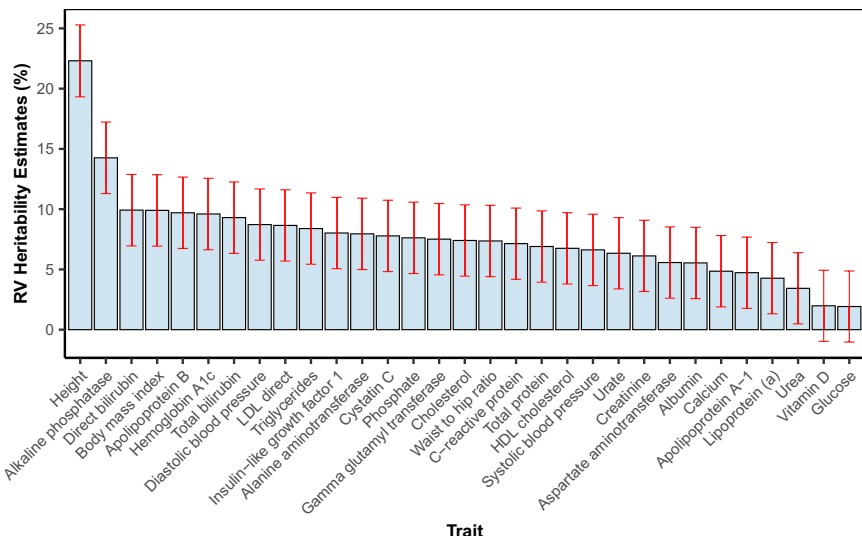

**Fig. 3 | Estimated phenotypic variance explained by RVs across 31 complex traits.** Bar chart illustrating the exome-wide RV heritability estimates ($h^2_{RV}$) +/− 95% confidence interval. Contribution of rare coding variants on 31 complex traits were based on $n = 167,348$ Caucasian individuals from the UK Biobank, as estimated using RARity via the exome-wide block construction. Traits were standardized for age, sex, and the first 20 genetic principal components. 95% confidence interval of trait $h^2_{RV}$ is denoted with red, vertical error bars.

extensive simulation studies and proved to produce robust estimates across a wide range of analytical scenarios.

The performance of RARity for CV contributions was comparable with BOLT. The estimated $h^2_{CV}$ by LDSC on the other hand was lower than RARity for each trait. While LDSC is a powerful tool that allows estimation $h^2_{CV}$ from GWAS summary-statistics, with adequate control for population stratification, it is prone to underestimation as discussed in several studies[28–31]. Comparison of the $h^2_{CV}$ for height and BMI, with other SNP-heritability methods in the literature, such as LDSC[23], LDAK[32], GCTA[33], GRE[19] and SumHer[34], shows that our estimates are on the higher end (but not the highest) of what has been historically reported in comparison studies[19,30]. The apparent higher $h^2_{CV}$ for height and BMI using RARity may be a characteristic of the subset of the population selected in this study. All current methods make assumptions about the genetic architecture, such as the effect size, variance, LD/MAF ratios, etc., since the true genetic architecture is unknown, it remains unclear which estimates in the literature are reliable. More importantly, none of these methods are suitable for estimation of rare variant heritability.

It was observed that for most traits, RVs account for a noticeable proportion of trait heritability independent of CV contribution. One of the technical challenges in combining CV and RV contribution is the presence of LD and its interplay with allele frequencies. The results suggested that the amount of LD between CVs and RVs is heterogeneous, and consequently, their independent contribution to heritability is trait dependent. Indeed, the "missing heritability" can be almost perfectly recapitulated by incorporating RVs for 11 of the 31 traits, including height and BMI (Supplementary Data 1), reaffirming the conclusion from a recent study that used whole genome sequencing (WGS) data[14]. Meanwhile, the apparent lack of recovery of the pedigree-based heritability for other traits may be due to the inherent characteristics of the pedigree-based studies. Generally, pedigree-based studies report higher heritability as compared to SNP-based heritability from population studies[21]. This gap is most likely due to limited sample size in the early genome-wide association studies, variability in sample collection, population characteristics, the exclusion of sex chromosomes[35], rare intronic, structural and non-coding regulatory variants, in addition to the non-additive effects which are captured by pedigree-based analysis but not necessarily captured by RARity, nor by the most commonly used CV heritability models.

Several methods have been used to increase the power of detecting gene-trait associations, the most popular methods being gene burden testing and Sequence Kernel Association Test (SKAT)[8]. Since SKAT aggregates the associations between variants and the phenotype through a kernel matrix[8] it is solely designed to test the strength of association via $p$-values without providing an effect size, and thus rendering it incomparable to RARity. The variant aggregation method used in gene-burden testing do not have such a limitation, and consequently, we were able to examine the effect of aggregating RVs on trait variance, which on average resulted in a 79.3% loss on the estimated $h^2_{RV}$. These results suggest that burden tests may have limited ability to predict traits. On the other hand, the use of unaggregated variants in RARity, not only captures more genetic variance, but also offers the practical advantage of characterizing genes based on $h^2_{RV-gene}$. Furthermore, with fewer variants per block, $h^2_{RV-gene}$ can be computed more efficiently to obtain the total heritability and still produce consistent results with those derived using larger blocks of exome data (Fig. 2b).

Next, we leveraged heritability estimates to assess the performance of RV pathogenicity algorithms in the context of complex traits. We see very modest, if any, enrichment of $h^2_{RV}$ when variants are filtered according to pathogenicity scores. This is likely because the pathogenicity scores for RVs currently used (MCAP, CADD, REVEL, etc.) are largely based on Mendelian diseases, have a modest impact on complex trait $h^2_{RV}$, and are limited in distinguishing the variants with any biological effects from those that do not. This affirms our conclusion that there is a need for alternative methods to study the functional consequences of RVs.

One of the useful features of RARity is that it can be tailored to identify genes significantly enriched for heritability and help with gene discovery and characterization. We show this by identifying 152 genes significantly enriched for heritability. To the best of our knowledge, 11 of these associations have not been previously described through exome wide association studies or GWAS (Table 2, Supplementary Data 2). We further demonstrate that the 152 genes with significant $h^2_{RV-gene}$ across 31 traits are enriched in various biological and disease pathways. For example, 11 significant genes contributing to heritability of ApoB are involved in pathways ranging from abnormal arterial stenosis to chylomicron, LDL and lipoprotein clearance, as well as hyperlipidemia and coronary artery diseases (Fig. 6, Supplementary

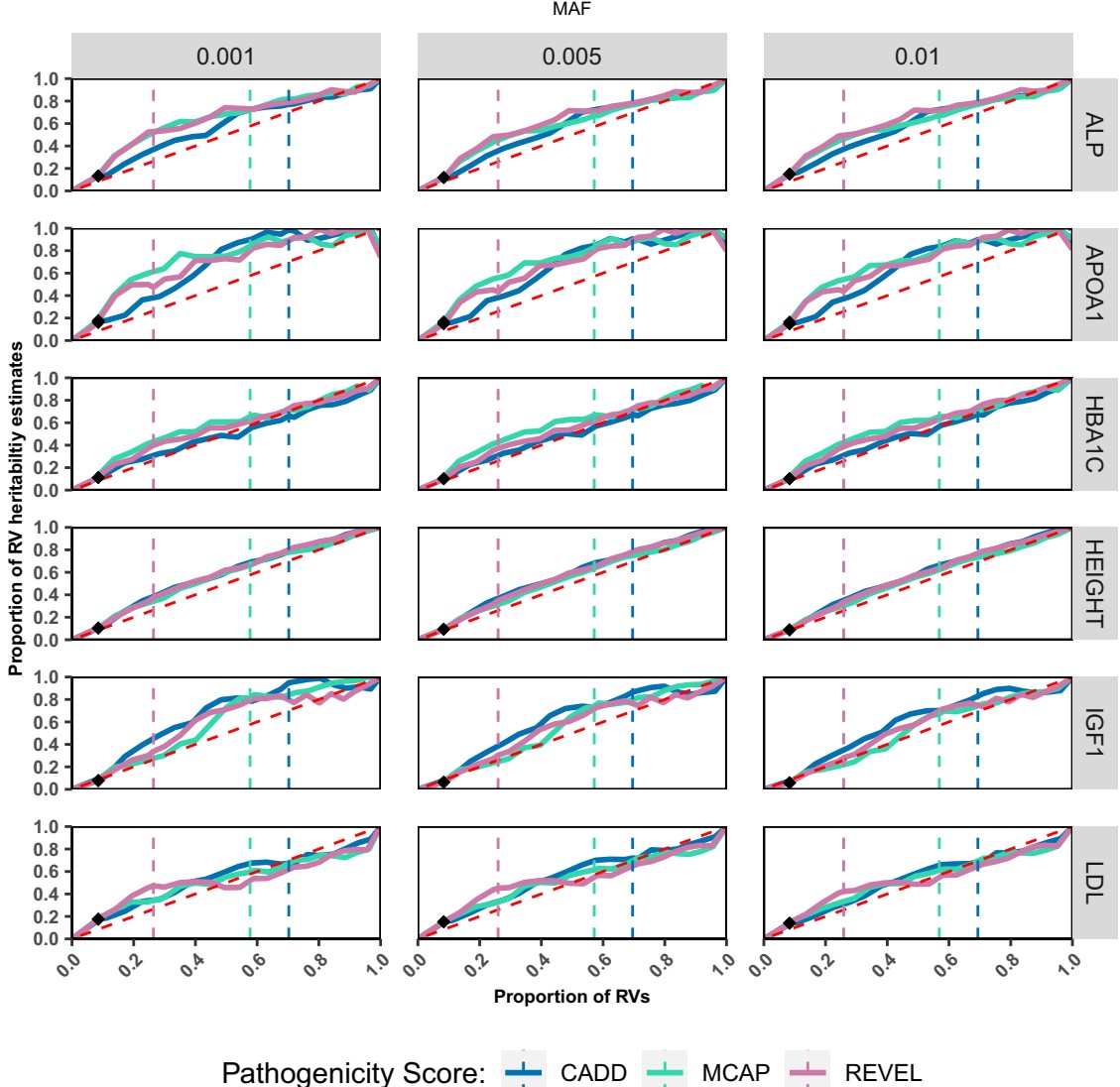

**Fig. 4 | Impact of pathogenicity scores on variance explained by RVs for 6 representative traits.** Plots illustrating the proportion of RV heritability explained (*y*-axis) as a function of incorporating increasingly "deleterious" genetic variants (*x*-axis). Proportion of heritability estimates is the fraction of estimates in relation to all protein altering and LoF variants within the MAF categories. The vertical, dashed lines represent the binary thresholds recommended to define pathogenicity for CADD (blue), M-CAP (green) and REVEL (magenta). The black diamond marks the point of last inclusion of LoF variants, which were prioritized before missense mutations. The diagonal dashed red lines represent the scenario wherein RVs uniformly contribute to $h^2_{RV}$, irrespective of pathogenicity score. ALP alkaline phosphatase, HBA1C hemoglobin A1c, LDL low density lipoprotein direct, IGF1 insulin line growth-factor 1, APOA1 apolipoprotein A-I.

Data 4 and 5). Interestingly, previous association studies did not identify the *PPARA* gene as genome-wide significant, even though *PPARA* significantly contributes to the $h^2_{RV}$ of ApoB and that both *PPARA* and *APOB* are involved in several lipid-related pathways (Supplementary Data 5), fatty liver disease, and dyslipidemias (Fig. 6), and identified as "druggable genome" in the DGIdb database (Supplementary Data 3). Notably, *PPARA* is also the target of the lipid-lowering drug class known as fibrates[36], making it a strong candidate for further pharmacogenomic studies. Indeed, most of the significant genes play important roles in disease etiology, as observed through DisGenet analyses. Further examples include the contribution of *TFAM* on alkaline phosphatase heritability and its role in hepatocerebral mitochondrial DNA depletion syndrome, and the contribution of *TMEM43* on HbA1c heritability and macular degeneration. These results imply that genes involved in diseases are also likely to contribute significantly to biomarker heritability. However, whether these associations can be used as markers of pathogenic mutations or represent causal mediations through biomarker concentrations will require further investigations.

As an application of the gene-wise blocks, we observed that for most traits, longer genes do not explain higher trait variance, this is likely because the relative paucity of larger genes obscuring any underlying relationship[16]. Our observations also show a modest increase in $h^2_{RV\text{-}gene}$ with higher RV evolutionary constraint (indicated by lower LOEUF scores[27]) for only a few traits, and from this, we can conclude that RVs contributing to the variance in complex traits are likely well-tolerated and are not selected against in general population.

This study has several limitations. First, exclusion of non-coding, singleton, and doubleton variants, as well as LD pruning of potentially functional variants, may lead to an underestimate of the $h^2_{RV}$. Although, numerous studies have shown that rare coding variants have major functional impacts in direct way[37], it is quite possible that some of the observed $h^2_{RV}$ is due to the coding RVs being in LD with the non-coding variants that are not represented in WES. On the other hand,

**Table 2 | Genes with significant heritability estimates for each trait**

| Trait | Genes with significant RV heritability estimates | Not reported in exome-wide associations in UKB** | Not reported in GWAS Catalogue |
|---|---|---|---|
| Albumin | ALB; FCGRT; TBC1D2B | | |
| Alkaline phosphatase | ALDH5A1; ALPL; ASGR1; GBGT1; GPLD1; HSPG2; NBPF3; TDP2; TFAM; ZNF800 | **TFAM** | **TFAM** |
| Alanine aminotransferase | GPT; MFSD3 | | MFSD3 |
| Apolipoprotein A-I | ABCA1; ADH1B; ANGPTL3; APOA1; APOC3; CETP; LCAT; LIPC; LIPG; LPL; PLA2G12A; SCARB1 | CETP | |
| Apolipoprotein B | APOB; APOE; BCAM; CEACAM20; CLASRP; LDLR; NECTIN2; NKPD1; PCSK9; PPARA; ZNF229 | **PPARA** | **PPARA** |
| Aspartate aminotransferase | ANO5; GOT1 | | |
| BMI | MC4R | | |
| Calcium | ALB; CASR; FCGRT | | |
| Cholesterol | ABCA1; ABCG5; ANGPTL3; APOB; FGB; JAK2; KHDRBS2; LDLR; LIPG; NECTIN2; NKPD1; PCSK9; ZNF229 | **KHDRBS2**; ZNF229 | **KHDRBS2** |
| Creatinine | LRP2; SLC22A2; SLC22A7 | | |
| C-reactive protein | ABCA1; APCS; CRP; JAK1; TMED8 | **TMED8** | **TMED8** |
| Cystatin C | CGNL1; CST3; SH2B3 | | |
| Direct bilirubin | ATG16L1; DGKD; DNAJB3; HJURP; MROH2A; NR1I2; SAG; SLCO1B1; SLCO1B3; SLCO1B3-SLCO1B7; TRPM8; UGT1A1; UGT1A10; UGT1A3; UGT1A4; UGT1A5; UGT1A6; UGT1A7; UGT1A8; UGT1A9; USP40 | NR1I2; SLCO1B3-SLCO1B7 | |
| Gamma glutamyl transferase | A1CF; GGT1; LRRC75B; RORC; SYNJ2 | | |
| Glucose | G6PC2; GCK | | |
| Hemoglobin A1c | ADGRE5; APEH; CTU2; G6PC2; GCK; JAK2; PFKL; PFKM; PIEZO1; RHAG; SPTB; TMC8; TMEM43 | **ADGRE5; TMEM43** | **ADGRE5**; APEH; JAK2;**TMEM43** |
| HDL cholesterol | ABCA1; APOA1; APOA5; APOC3; CETP; LCAT; LIPC; LIPG; LPL; NR1H3; PLA2G12A; SCARB1 | | |
| Height | CRISPLD2; DDR2; FGF2; FGFR3; GH1; GHRH; GRAMD2A; HAPLN3; IHH; NPR2; NPR3; SCMH1; STC2; ZFAT | GH1; GHRH | |
| IGF-1 | GH1; IGFALS; IGFBP3; PARPBP; ZNF12 | | |
| Low-density lipoprotein | ABCG5; ANGPTL3; APOB; APOE; LDLR; NECTIN2; NKPD1; PCSK9; ZNF229 | | |
| Lipoprotein(a) | ACAT2; AGPAT4; ARID1B; EZR; FNDC1; IGF2R; LPA; MAP3K4; MAS1; MRPL18; PLG; PNLDC1; SLC22A1; SLC22A2; SLC22A3; SOD2; SYNJ2; SYTL3; TMEM181; TULP4; WTAP | ACAT2; ARID1B;**EZR**; SYNJ2; TULP4 | **EZR**; TMEM181 |
| Phosphate | ALPL; CDR2; ENPP1; HLA-DPA1; TTK | **CDR2; HLA-DPA1; TTK** | **CDR2; HLA-DPA1; TTK** |
| Total bilirubin | ATG16L1; DGKD; DNAJB3; HJURP; MROH2A; NR1I2; SAG; SLCO1B1; SLCO1B3; SLCO1B3-SLCO1B7; TRPM8; UGT1A1; UGT1A10; UGT1A3; UGT1A4; UGT1A5; UGT1A6; UGT1A7; UGT1A8; UGT1A9; USP40 | **NR1I2**; SLCO1B3-SLCO1B7 | **NR1I2** |
| Total protein | FCGR2B; FCGRT; SNX8; TNFRSF13B | | |
| Triglycerides | ANGPTL3; APOA5; APOB; APOC3; LPL; PLA2G12A; SIK3; ZPR1 | | |
| Urate | LGALS13; PDZK1; SLC22A11; SLC22A12; SLC2A9; WDR1 | | |
| Urea | RBM47; CYP2R1; HAL; PDE3B | | |
| Vitamin D | CYP2R1; HAL; PDE3B | | |

Systolic blood pressure, diastolic blood pressure and waist-to-hip ratio did not have significant $h^2_{RV-gene}$. Bolded font indicates gene-trait relationships unidentified in previous studies. ** Wang, Q. et al. [64] and Backman, J.D. et al. [65].

very little is known about the rare non-coding regions, which are difficult to define and even more difficult to assess functionality[38], consequently there is no direct comparison of the contribution of rare-coding vs. non-coding variants in the literature. This is indeed a question of high importance and may be answered with WGS. We anticipate that RARity can be applied to WGS with further calibrations, to study effects of other variants not discussed here. One of the challenges of WGS data analysis is that there is no natural biological unit available in the intergenic regions. With RARity, there is no need for defining the biological units, as the estimates can be based on blocks that are agnostic of genetic borders.

Currently, the RARity model is fitted for to continuous traits. For dichotomous trait heritability, the model will require slight adjustments to transform the observed heritability to a liability scale heritability to account for the case-control imbalance in population study[39]. We did not find any significant difference in the RV heritability between males and females across the 31 quantitative traits studied (Supplementary Fig. 4). In contrast, SNP-based studies showed slight differences in heritability of selected traits between the sexes[40–42]. Since the analysis using RVs was limited by the reduced power, further research of larger sample sizes is required to better understand the sex effect on RV heritability. We have only explored a fraction of the pathogenicity scores and further research will benefit from using RARity as a tool to evaluate the algorithms to distinguish functional vs. non-functional variants. In addition, heritability describes the phenotypic variance at the population level, and thus cannot be generalized to a different population, and nor does it inform the prediction of phenotypic variations of an individual. Therefore, methods to estimate

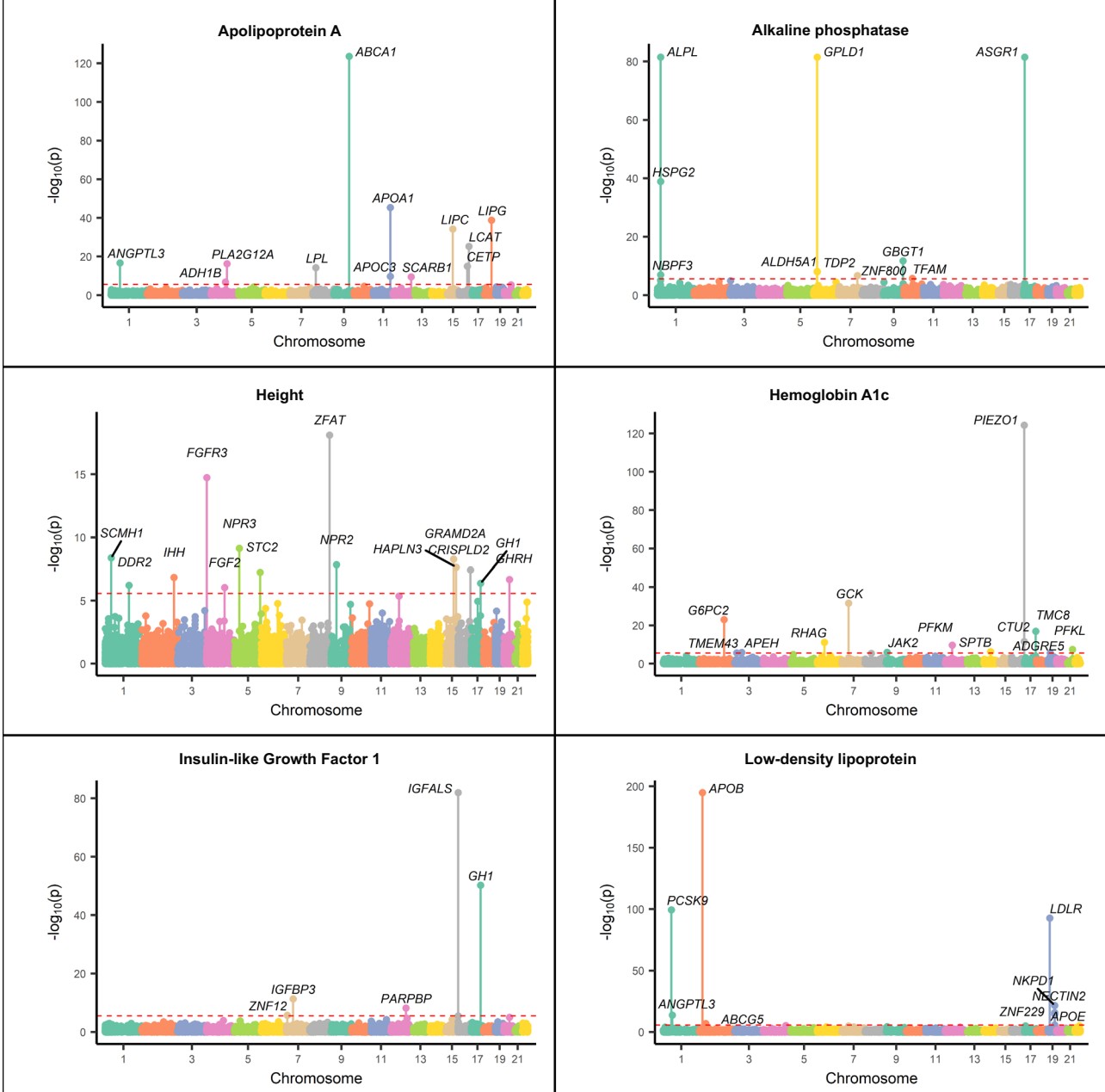

**Fig. 5 | Manhattan plot illustrating genes with significant heritability for selected traits.** RARity was used to determine $h^2_{RV\text{-}gene}$ for all genes with qualifying RVs. Each dot represents a single gene, with genes ordered on the *x*-axis according to their genomic position. The *y*-axis represents the significance of $h^2_{RV\text{-}gene}$ measured as -$\log_{10}$ transformed *p*-values, where the *p*-values were derived using *F*-test. Red, horizontal dashed lines mark the Bonferroni's *p*-value significance threshold corrected for 18,214 genes (*p*-value < $2.75 \times 10^{-6}$). Genes with significant $h^2_{RV\text{-}gene}$ are labelled.

RV heritability, such as RARity, are not intended to replace RV association methods, but rather to complement.

Together, these results confirmed that (1) RVs can account for a significant portion of the complex trait heritability, (2) gene-level RV aggregation (gene burden) leads to a substantial loss of information, (3) identification of genes significantly enriched for $h^2_{RV\text{-}gene}$ can help with gene discovery, and (4) innovative methods are needed to predict variant-level functionality. In conclusion, the high trait variance explained by RVs makes it imperative to continue to invest in the study of RVs and understand their impact on health and diseases. As such, future studies extending the methodology to analysis of dichotomous traits, particularly disease status, are in a pressing need.

## Methods
### Study population
The UK Biobank (UKB) study is a prospective cohort comprising of approximately 500,000 participants (ages 40–69 years) with extensive genotypic and phenotypic data from consenting individuals[43]. All UKB data included in our analyses were accessed as part of our approved application #15255. Here we utilized two main genetic datasets from the UK Biobank. First, our primary source of RVs was the WES data with 17,975,236 variants on 200,643 participants (UKB data field: 23155). Second, CVs were extracted from imputed genotype data on 488,264 individuals (data field: 22418; v3 release). The acquisition and primary quality control (QC) of both genetic data are described elsewhere[44]. Briefly, out of the 200,643 samples with WES data,

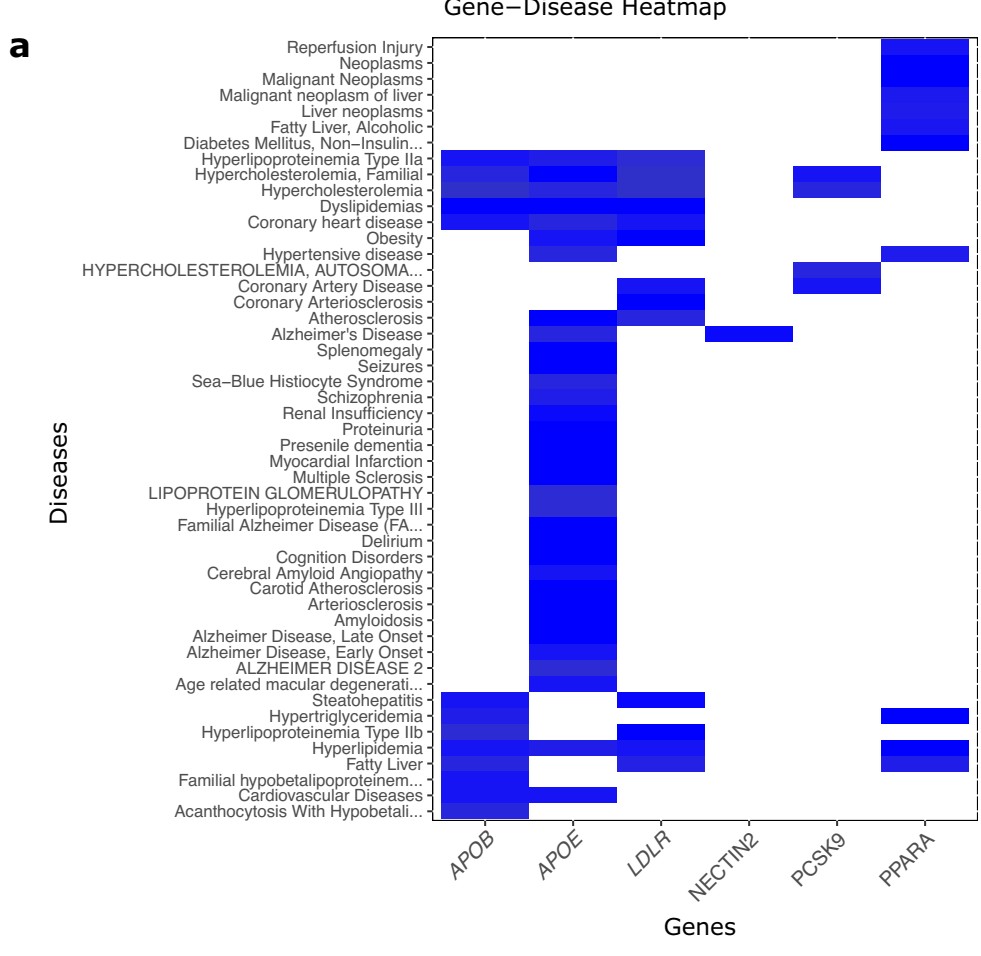

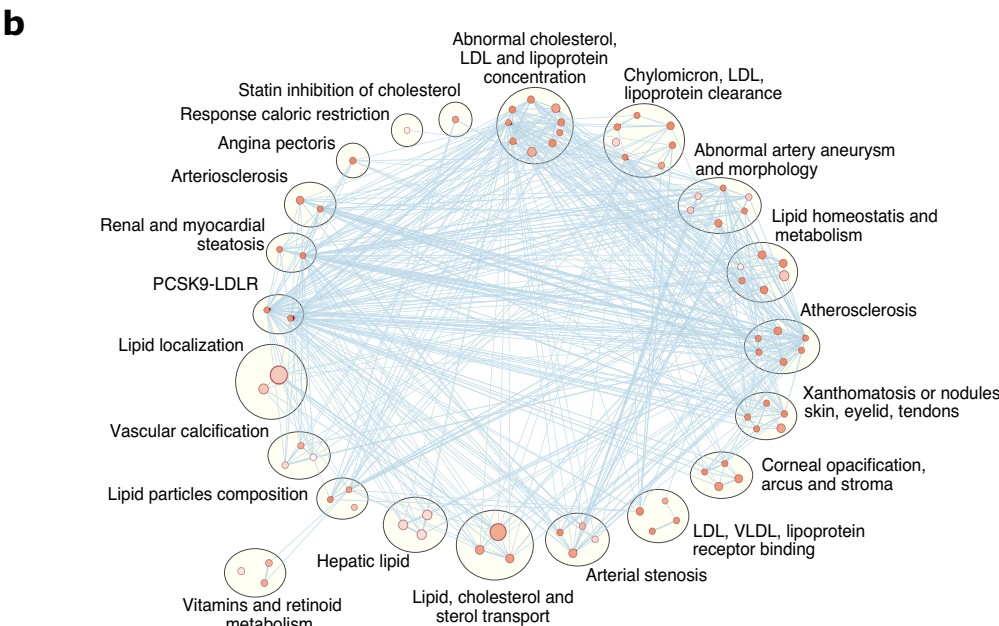

**Fig. 6 | Network and pathway analyses of genes contributing significantly to ApoB. a** Gene-disease heatmap, created with DisGeNET, for the genes contributing significantly to the RV heritability of ApoB. The intensity of colour is proportional to the strength of evidence for gene-disease-association, with darker colour representing a greater level of evidence. **b** Network cluster representing biological mechanisms influenced by genes contributing significantly to ApoB heritability estimates. The nodes represent gene-sets, the edges represent known interaction between biological pathways. Node size reflects the number of genes in the gene-set. More intense colour indicates higher enrichment value.

individuals were excluded based on: consent withdrawal ($n = 11$), call rates less than 99% ($n = 2$), discordance between genetic and reported sex ($n = 18$), a departure from putative ancestorial clusters based on the first two genetic principal components ($n = 3$), assigned cluster membership to a continental population with less than 5000 samples ($n = 12,765$, of which, South Asian = 3395; African = 3168; Other = 6202), and 3rd degree or closer relatedness ($n = 14,156$). In the remaining 173,688 individuals, an additional 6340 were removed following QC of biomarker data (as described below). We focused on the 167,348 unrelated Caucasian participants to estimate narrow-sense heritability contributed by CVs, RVs, or the combined CV and RVs.

## Biomarker and anthropometric data

The blood biomarkers in UKB represent clinical diagnostic measures and established risk factors for diseases. For example, HbA1c is used in the diagnosis of diabetes and lipids are needed for risk stratification of cardiovascular diseases. Besides the standard QC steps implemented by the UKB study team[45], we applied additional steps to curate the final list of 26 biomarkers and 5 anthropometric traits (height, body mass index [BMI], waist-to-hip ratio, and systolic and diastolic blood pressure measured automatically) (Supplementary Table 1, Supplementary Fig. 1). Briefly, sex-specific biomarkers, such as sex hormone binding globulin (SHBG) and testosterone, as well as those biomarkers with >80% missingness (e.g. oestradiol and rheumatoid factors) were excluded from the analyses. Next, we winsorized values that were either above or below the detectable range, using the reportability fields for each biomarker. Out of the 173,688 participants with WES data, we removed 598 individuals with missing values for all 26 biomarkers. Since many of the examined biomarkers can be altered by specific medications including blood glucose, HbA1c, lipids, and blood pressure, we applied corrections for medication status (Supplementary Table 2). As a result, 5,742 individuals on glucose-lowering drugs were removed, meanwhile, individuals on statins had their baseline low-density lipoprotein cholesterol (LDL-C) and ApoB values adjusted by dividing by 0.7, their total cholesterol adjusted by dividing by 0.8, and their ApoA-I and high-density lipoprotein (HDL) cholesterol adjusted by dividing by 1.06 and 1.05, respectively[46–49]. Blood pressure-lowering medications were adjusted by adding 10 mmHg and 15 mmHg to their diastolic blood pressure (DBP) and systolic blood pressure (SBP), respectively[50,51]. Further, missing biomarker or trait values were imputed by their mean values. All traits were then quantile transformed to resemble a standard normal distribution and further adjusted for age, sex, and the first 20 genetic principal components (PCs) to account for any effects of sub-population structure within UKB[52], and finally standardized to have mean 0 and variance 1. Biomarker treatment for sex-stratified analysis was performed in an identical manner.

## Genotype data quality control

**Rare coding variants.** Genetic variants were called from WES data following the Functional Equivalent pipeline[53]. All monomorphic variants ($m = 83,700$), variants with missing genotypes in more than 10% samples ($m = 369,215$), and those deviating significantly from Hardy-Weinberg Equilibrium ($p$-value $< 5 \times 10^{-6}$; $m = 35,317$) were removed. Remaining variants were annotated with predicted pathogenicity scores, and amino-acid changes using ANNOVAR *geneanno* pipeline with the refGene database[54]. Variants were annotated with MAF based on the UKB samples, as well as the five major ancestries identified in the Genome aggregation database (gnomAD 2.11): Latino, non-Finnish European, African/African American, East Asian, South Asian[27]. Qualifying RVs were defined as variants that were nonsynonymous single nucleotide variants, frameshift deletions or insertions, in-frame deletions or insertions, stop-gain, stop-loss and start-loss variants, with a minor allele count (MAC, the number of minor alleles at each locus in

the population being studied) >2 and MAF below the cut-off (<1%, <0.5% or <0.1%) in all gnomAD subpopulations, and locally in the UKB samples. Within these variants, the stop gain/loss variants and frameshift variants were defined as the LoF variants, and the rest are referred to as protein-altering variants. RVs were also subset into MAF bins (MAF = 0.01–0.005, 0.005–0.001 or 0.01–0.001) to examine the contribution of different MAF categories to $h^2_{RV}$.

To reduce the influence of long-range LD between variants that would otherwise inflate the overall heritability estimate, as we show in subsequent simulations ("Calibrating RARity"), highly correlated RVs were removed using PLINK1.9[55], by LD pruning with a Pearson's $r^2$ threshold > 0.1 within a window of 50 Mb that was shifted by 500 bases at the end of each step. Next, individuals on glucose-lowering medications were removed and a MAC filter was applied once more to retain RVs with MAC > 2, leading to a final analytical dataset including 167,348 individuals and 1,592,257 variants with MAF < 1% in 18,213 genes. All other analyses were based on this analytical dataset. Individual level genotypes were extracted with PLINK1.9, assuming an additive model for all variants, and thus allotting a score of 2 for rare allele homozygous variants, 1 for heterozygous variants, and 0 otherwise. Multi-allelic variants were treated as bi-allelic by considering the presence or absence of (any) rare variant. While RVs with missing genotype in >10% samples were removed during the early QC steps, mean imputation was employed to fill in the missing genotypes in the remaining samples. Finally, genotypes were standardized to have mean 0 and variance 1. The genotype and phenotype data processing steps are illustrated schematically in Supplementary Fig. 1.

**Common variants.** CVs originated from the third release of the UK Biobank genotype data in 2017. The imputed genotypes are based on the Genome Reference Consortium Human Build 37 (GRCH37), and further filtered to retain CVs with imputation quality score greater than 0.7, those with no significant deviation from Hardy-Weinberg equilibrium ($p$-value $> 1 \times 10^{-10}$). We further filtered genotype data by LD pruning with $r^2 > 0.9$ and a rolling window of 1 Mb, that were shifted in steps of 500 bases, leaving 1,030,594 CVs in 159,058 participants, for whom we also had WES data. These pruning parameters were selected based on simulations using a very similar approach, using 325,989 participants in the UKB by Di Scipio and Khan et al.[31]. For compatibility with RVs, the CVs were lifted to the GRCH38 assembly using UCSC LiftOver[56]. Similar to RVs, individual level genotypes were extracted with PLINK1.9, assuming an additive model, mean imputation was also employed to fill in the missing genotypes (in <10% samples), followed by standardized to have mean 0 and variance 1.

**Combined common and rare coding variants.** We concatenated the derived RVs and CVs, and then LD pruned once again with the same parameters as CVs ($r^2 > 0.9$, window size = 1 Mb, step size = 500 bases). We already implemented a more stringent LD pruning schema for RVs ($r^2 > 0.1$, window size = 50 Mb, step size = 500 bases), conversely, an overly stringent LD pruning schema applied to CVs would remove most of the variants, and thus, for the combined CV + RV we implemented the more relaxed LD pruning similar to that for CV ($r^2 > 0.9$, window size = 1 Mb, step size = 500 bases) and blocks of 20,000 variants were constructed to enable distribution of both CVs and RVs within each block.

**Statistical model to estimate RV heritability using RARity**

We developed a method, RARity, to compute heritability estimates based on aggregating linear regression models over large genetic regions including up to thousands of variants. The method is based on a multivariate linear regression model:

$$Y = G\beta_G + \epsilon, \tag{1}$$

The RARity method entails computing the multiple linear regression solutions using an ordinary least square (OLS) for each of the non-overlapping genetic blocks $(1,\ldots,k,\ldots,K)$ in parallel under the condition that $n$ is much larger than the number of genetic variants $(p_k)$ in the $k$ th block, while ensuring the between block correlation, due to linkage disequilibrium (LD) spillage between blocks, is minimized. In other words, RARity approximates the linear solution to $\boldsymbol{\beta_G}$ by setting the observed $(\boldsymbol{G'G}) \in R^{m \times m}$ to a block diagonal matrix, where $m = \sum p_k \gg n$. Given the observed quantitative trait $y$, the OLS estimate of the genetic effects vector for the $k$ th block $\boldsymbol{G_k}$ is:

$$\hat{\boldsymbol{\beta}}_{\boldsymbol{G_k}} = (\boldsymbol{G'_k G_k})^{-1} \boldsymbol{G'_k y}, \tag{2}$$

and the fitted value, denoted by $\hat{y}$, can be computed as:

$$\hat{\boldsymbol{y}} = \boldsymbol{G_k} \hat{\boldsymbol{\beta}}_{\boldsymbol{G_k}}. \tag{3}$$

The estimated heritability associated with the $G$ matrix is simply the amount of variance explained by the fitted value:

$$R^2_{G_k} = \hat{\boldsymbol{y}}' \hat{\boldsymbol{y}} / \boldsymbol{y}' \boldsymbol{y}. \tag{4}$$

Since $R^2_{G_k}$ increases as the number of predictors increase, the adjusted $R^2$, denoted by $\bar{R}^2$, is used as our estimate for the proportion of variance explained. Total RV heritability of the trait is estimated by the sum of $\bar{R}_k^2$ over all K blocks:

$$\hat{h}^2 = \sum_{k=1}^{K} \bar{R}^2_{G_k} = \sum_{k=1}^{K} \left[ 1 - \left( 1 - R^2_{G_k} \right)(n-1)/(n-p_k-1) \right]. \tag{5}$$

The 95% confidence interval (CI) of $R^2_{G_k}$ can be approximated for each block using the asymptotic properties described by Algina[57] using the Wald's method, where the variance of $R^2_{G_k}$ is given by:

$$\widehat{Var}\left(R^2_{G_k}\right) = \frac{4R^2_{G_k}\left(1 - R^2_{G_k}\right)^2 (n-k-1)^2}{(n^2-1)(n+3)} \tag{6}$$

The 95% CI for the adjusted $R^2$ of a single block can then be derived accordingly:

$$\bar{R}^2_{G_k} \pm 1.96 \sqrt{\widehat{Var}\left(\bar{R}^2_{G_k}\right)}, \tag{7}$$

where

$$\widehat{Var}\left(\bar{R}^2_{G_k}\right) = \left(\frac{n-1}{n-p_k-1}\right)^2 \widehat{Var}(R^2_{G_k}). \tag{8}$$

To estimate the 95% CI for $\hat{h}^2$, we approximated the asymptotic variance by the sum of the individual sampling variance, $\widehat{Var}(\bar{R}^2_{G_k})$ for each block, assuming each block is roughly uncorrelated of others after controlling for LD spillage:

$$\widehat{Var}\left(\hat{h}^2\right) = \sum_{k=1}^{K} \left(\frac{n-1}{n-p_k-1}\right)^2 \widehat{Var}\left(R^2_{G_i}\right), \tag{9}$$

which then translates to a Wald's 95% CI for the $\hat{h}^2$:

$$\hat{h}^2 \pm 1.96 \sqrt{\widehat{Var}\left(\hat{h}^2\right)}. \tag{10}$$

Since the model is conditional on the observed genotype matrix, RARity requires no parametrization nor assumptions regarding the genetic architecture of traits analyzed (such as polygenicity of effects or relationships between MAF/LD and effect size). The only

assumption of block-wise independence was addressed by a stringent LD pruning to avoid long-range LD. The main computational burden for biobank scale datasets is the inversion of the $\boldsymbol{G'G}$ matrix, which is of size $m \times m$, but the matrix calculation becomes quite manageable for blocks of $\boldsymbol{G}$, where each block contains $p_k = 5000-10,000$ variants and $n \sim 200,000$ in a standard high-performance computing environment. For example, for a typical analysis of 5000 RVs and 167,348 samples, the heritability estimation took approximately 7 minutes for all 31 traits, running on a single core and 7.3 Gb memory. All statistical analyses were performed using the statistical programming language R (version 3.6.0)[58].

## Statistical power

Statistical power of $h^2_{RV}$ was estimated empirically from the variance of 10,000 simulated $h^2_{RV}$ $(\widehat{Var}(\hat{h}^2))$, under 230 conditions of sample sizes and true set $h^2_{RV}$. Non-central $F$-distributions were used to simulate the observed genetic effects at each genotype block, and exome-wide $h^2_{RV}$ was derived as described above. The true set $h^2_{RV}$ ranged from 2 to 25%, with increments of 5%. Sample size was varied from 25,000 to 250,000 individuals by increments of 10,000. For each condition, the statistical power was calculated as the proportion of observed $p$-values less than 0.05 out of the 10,000 simulations.

## Calibrating RARity

Rarity can be used with both common and rare variants. The effect of LD on over-estimation of heritability is well-known for common variants[20] and can be minimized by choosing a suitable set of pruning parameters. In our preliminary analyses, we also observed over-estimation of $h^2_{RV}$, which occurs when rare variants in different blocks are in LD, thus, it was necessary to perform pruning and calibrate the pruning parameters to empirically minimize bias.

Here we conducted simulation studies using observed genotypes to identify the most suitable LD pruning parameters, including the Pearson's $r^2$ coefficient threshold, the window size, and the step size, to reduce bias in estimating heritability. To this end, RVs on each chromosome, with MAF < 0.01, were LD pruned under 7 different scenarios that varied in window sizes ($ws$) and LD $r^2$ threshold in the following combinations ($ws = 1$ Mb with $r^2 > 0.9$, 0.5 or 0.1; $ws = 20$ Mb with $r^2 > 0.1$; $ws = 50$ with $r^2 > 0.9$, 0.5 or 0.1), with a fixed step size of 500 bases. Pruned variants in each scenario were then partitioned into exome-wide blocks, with 5000 variants per block. We assumed that a random subset of 20% of the RVs in each block had an independent effect associated with the simulated trait of interest and their contribution to true set $h^2_{RV}$ was 0.05 for the whole exome. The unobserved genetic effects were simulated from a standard normal distribution while the errors were sampled independently from a normal distribution with mean 0 and variance 0.95.

The simulated phenotype ($\mathbf{Y_{sim}}$) was then computed as:

$$\mathbf{Y_{sim}} = \boldsymbol{G\beta_G} + \boldsymbol{\epsilon} \tag{11}$$

We generated 20 phenotypes ($\mathbf{Y_{sim}}$) under each of the 7 scenarios to assess the overall impact of LD pruning parameters on RARity estimates and proceeded to estimating exome-wide $h^2_{RV}$ as described in the section "Statistical model to estimate RV heritability using RARity". In addition, the calibration was repeated with the assumption that the true $h^2_{RV}$ originated from 10% of RVs within 5% of the genes, instead of homogeneous distribution of $h^2_{RV}$ across all blocks.

To test whether the more relaxed LD pruning for CVs ($r^2 > 0.9$, $ws = 1$ Mb, step size = 500 bases) would impact the estimation of $h^2_{CV}$, we benchmarked RARity against alternative methods designed for CVs, namely BOLT[22] and LDSC[23].

## Testing the effects of genetic architecture

To assess whether RARity is sensitive to MAF thresholds, values of true $h^2_{RV}$, or varying fractions of causal variants or genes, we further tested RARity with simulations utilizing real genotype data, pruned with the default LD pruning threshold ($r^2 > 0.1$ within a window size of 50 Mb). For each simulation, exome-wide $h^2_{RV}$ was estimated for $Y_{sim}$. A total of 20 simulations were carried out for each scenario as follows. The effect of varying MAF was tested by assuming that 10% of variants within 5% of genes are causal in each of the MAF categories (MAF < 0.001, <0.005 and <0.01). The effect of varying $h^2_{RV}$ was tested for MAF < 0.01 with the true $h^2_{RV}$ set to either 2, 5, 10, 20 or 25%. Next, keeping a consistent MAF < 0.01 and true $h^2_{RV}$ at 10%, the effect of varying fractions of causal genes (0.01, 0.05, 0.1 and 0.2) was tested with 10% of RVs with true effects. Finally, keeping a consistent MAF < 0.01, and true set $h^2_{RV}$ at 10%, we tested the effect of varying fractions of causal variants (0.001, 0.05, 0.1 and 0.2) in 5% of randomly selected genes.

## Application to UKB data

We estimated RV, CV ($h^2_{CV}$) and combined CV and RV heritability ($h^2_{tot}$) of 31 complex traits, including 26 biomarkers and 5 anthropometric traits in 167,348 UKB samples. For RV heritability, we additionally examined the influence of block construction on the final heritability estimate. The qualifying RVs were arranged in three different ways to create blocks of genotype data that were used to estimate heritability: (1) gene-burden blocks were derived by summing the number of rare alleles within each gene for an individual, which produced a single block containing all gene-burden scores as predictors. Heritability estimates from this type of block is denoted with $h^2_{RV\text{-}burden}$; (2) gene-wise blocks, consisted of un-aggregated RVs partitioned by gene, such that each block contained all the variants within a single gene. In other words, there were as many blocks as the number of genes. Gene heritability from this type of block is referred to as $h^2_{RV\text{-}gene}$, and the total trait heritability based on all genes is denoted as $h^2_{RV\text{-}gene\text{-}tot}$; (3) exome-wide blocks, were created by partitioning RVs in each chromosome into blocks of ~5000 adjacent RVs. This type of construct results in blocks that are gene-agnostic, i.e., independent of gene borders with varying number of genes per block, for example, the number of genes ranged from 17–170 per exome-wide block when protein altering and LoF variants (<0.01 MAF) were selected. The total heritability estimates from exome-wide blocks are denoted with $h^2_{RV}$. In each case, LD spillage was minimized through pruning prior to creating the blocks (as described above, in the "Genotype data quality control" section).

The impact of block size on $h^2_{RV}$ was tested using the first 4 blocks of ~5000 protein altering and LoF RVs (MAF < 0.01) from chromosome 22, where 2 consecutive blocks were combined to create blocks of ~10,000 RVs and all four blocks were combined to form a single block of ~20,000 RVs. Since the size of blocks did not impact the accuracy of $h^2_{RV}$ (Supplementary Fig. 11), the choice of 5000 RVs per block in an exome-wide block construct was motivated by computational efficiency.

## Enrichment analysis (pathogenicity and gene set)

**Identification of significant genes.** For each trait, we assessed 18,214 genes for their contribution to the total heritability and prioritized those with significant contribution for functional enrichment. A statistically significant contribution was determined by an F-test for regression models, against the null hypothesis that the gene-level heritability was zero, at a Bonferroni corrected gene-wide significance threshold of $\alpha = 0.05/18{,}214 = 2.75 \times 10^{-6}$.

**Pathogenicity scores.** We evaluated three well-known pathogenicity scores to classify deleterious RVs[59], namely, Combined Annotation Dependent Depletion (CADD) scores (v.1.6)[6], Mendelian Clinically Applicable Pathogenicity (M-CAP 1.3) scores[60], and rare exome variant ensemble learner (REVEL)[5]. We created subsets of RVs based on the default thresholds indicative of deleteriousness (CADD > 20, M-CAP > 0.025, and REVEL > 0.5[5,6,60]), as well as decreasing proportion of variants, by increasing the pathogenicity scores every 5th percentile. This was repeated for all three MAF cut-offs (<1%, <0.5% or <0.1%), and thus creating 63 subsets of RVs (21 pathogenicity cut-offs × 3 MAF thresholds) for each class of pathogenicity score. To derive an independent set of deleterious variants, we applied LD-clumping (Pearson's $r^2 > 0.1$, window size = 50 Mb) within each subset and retained the more pathogenic, independent variants with the highest score. We then constructed gene-wise blocks and derived $h^2_{RV\text{-}gene\text{-}tot}$, to efficiently estimate the total heritability for each subset of RVs. The proportion of $h^2_{RV\text{-}gene\text{-}tot}$ was measured in relation to all protein altering and LoF variants within each MAF categories. The proportion of $h^2_{RV\text{-}gene\text{-}tot}$ explained as a function of incorporating increasingly "deleterious" genetic variants was used to measure the performance of pathogenicity scores to identify sets of functional RVs.

**Network and pathway analyses.** The g:ProfileR[61] web tool for functional profiling, g:GOST, was used to test the enrichment of the genes with significant heritability ($h^2_{RV\text{-}gene}$ $p$-value < $2.75 \times 10^{-6}$), against gene-sets in common databases. The significant heritability genes for each trait were treated as separate gene lists for independent query, and statistical tests were conducted within a domain scope of only annotated genes, considering the GO biological process, GO molecular function, GO cellular component, KEGG, Reactome, TRANSFAC, miRNA, CORUM, HP, HPA, and WikiPathways data sources, and removing electronic GO annotations. This analysis resulted in a list of statistically significant enriched terms for each gene list, adjusted for multiple testing using g:SCS (set counts and sizes) $p$-value < 0.05, integral to the g:Profiler server. An Enrichment Map was then created with the results from g:Profiler, using Cytoscape version 3.10.1[62], with an FDR $q$-value cut-off value set to 0.001, and medium connectivity, producing networks where the nodes represent enriched pathways, and the edges represent all pairwise connections. These threshold values were chosen to ensure that we captured highly enriched pathways while being able to observe structure in the network. Next, we clustered the nodes to distinguish the major theme in the enriched pathways for each trait. DisGeNET[25] was used to retrieve and explore the gene-disease associations for the significant list of genes, resulting in a disease-heat-map for each trait. In addition, we ran a query of the target genes in the Drug-Gene Interaction database (DGIdb)[26] to identify potentially druggable and clinically actionable genes.

## Heritability estimates in relation to sex and gene-level annotations

Sex-stratified analyses were performed using gene-wise blocks including RVs with MAF < 0.01, including 92,963 females and 74,385 males. Differences in the heritability estimates between the genetic sexes was determined with $t$-test. In order to investigate the influence of gene length and evolutionary constraint on RV heritability, we tested the association between these independent predictors and heritability contribution from each gene towards RV trait $h^2$ ($h^2_{RV\text{-}gene}$). The predictors included (1) gene-level evolutionary constraint, which was determined using gnomAD pLoF Metrics (v2.1.1)[27]; and (2) gene length, which was based on the curated RefSeq transcripts obtained via the UCSC genome browser. For each gene, the transcript with the longest transcript length was selected to represent the length of the gene. Since the distribution of the gene lengths are skewed, with some genes being much longer than others, the lengths of all genes were log transformed. Heritability estimates for both analyses were calculated using the same set of 1,592,257 RVs with MAF < 1%.

To examine the enrichment of $h^2_{RV}$ in highly conserved regions, we estimated the $h^2_{RV\text{-}gene\text{-}tot}$ of height originating from the conserved

gene clusters, such as the hox, histone, protocadherin, and the hemoglobin gene clusters. The list of genes in these clusters was obtained from the HUGO Gene Nomenclature Committee (HGCN) database (www.genenames.org) and cross-listed with the genes carrying RVs in the UKB. The impact of gene orientation on $h^2_{RV}$ was examined by comparing the $h^2_{RV\text{-gene-tot}}$ contribution of the genes transcribed from the positive vs. negative strand, where the strand of each gene was annotated using the RefGene database in the UCSC genome browser[56], with roughly equal number of genes present on either strand.

### Reporting summary

Further information on research design is available in the Nature Portfolio Reporting Summary linked to this article.

## Data availability

Individual genetic and phenotypic data were obtained from the UK Biobank (http://www.ukbiobank.ac.uk/), under application #15255. The UK Biobank study received approval from the National Health Service National Research Ethics Service North West. Access to the UK Biobank individual-level data is not publicly available and must be obtained via an application (https://www.ukbiobank.ac.uk/register-apply/). UCSC genome browser (https://genome.ucsc.edu/) was utilized to access LiftOver for the conversion of Genome Reference Consortium Human Build 37 (GRCH37) to Genome Reference Consortium Human Build 38 built, and to obtain additional gene-level annotations such as gene-length, strand orientation and the gnomAD pLoF Metrics. Databases for gene-disease associations (DisGeNet, https://www.disgenet.org/), Drug Gene Interaction database (DGIdb, version 4.2.0, https://www.dgidb.org/) and HUGO Gene Nomenclature Committee (HGCN, https://www.genenames.org/) were utilized to inform on the importance of the target genes. Variant level annotations and pathogenicity scores, such as Mendelian Clinically Applicable Pathogenicity (M-CAP) Score, and Rare exome variant ensemble learner (REVEL) were obtained using ANNOVAR or downloaded directly from the web-based platform, such as the Combined Annotation Dependent Depletion (CADD) scores. Source data for all main and supplemental figures are provided with this paper.

## Code availability

All custom code and relevant documentation to run RARity is available in a public GitHub repository (https://zenodo.org/records/10426710)[63].

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

## Acknowledgements

The authors are thankful or all the funders, participants, and investigators of the UK Biobank. All UK Biobank data included in our analyses were accessed as part of our approved application #15255. We would also like to thank Dr. Andrew McArthur and his laboratory at McMaster University for providing GPU support.

## Author contributions

N.P. conceived and implemented the project, performed computational analyses, analyzed results, and wrote the manuscript. G.P. conceived and supervised the overall project. W.D. and M.K. provided expert guidance and feedback on statistical analyses and writing. M.D.S. and S.M. performed the CV related computational analyses. M.P. and R.L. curated the list of medications for diabetes and lipids. M.R.C. performed primary QC of genotype data and R.W.M. created Fig. 1. All authors reviewed and edited the manuscript.

## Competing interests

The authors declare no competing interests.
