## [Peer Review File · Nature Communications]

A Method to Estimate the Contribution of Rare Coding Variants to Complex Trait HeritabilityREVIEWER COMMENTS

Reviewer #1 (Remarks to the Author):

Estimating heritability without pedigrees is challenging. The authors provide an interesting and useful new method called RaRity for estimating heritability with rare variants for complex traits in large cohorts. In addition to aggregated RV count per gene, they proposed two frameworks to use unaggregated RVs either by sliding window of 5000 RVs or variable windows of RVs defined by genes. They also compared the RV based heritability with common variants (CV) based heritability in the UK Biobank cohorts with dozens of complex traits. They also showed the utility of this new measurement in the novel gene discoveries. Overall, the method is described well, and the results are compelling. Here are some concerns and suggestions for consideration to improve the clarity of the paper.

Major concerns

1. All studied individuals in this study are unrelated Caucasian participants. Please show the PCA plots to see if there are sub-structures of the populations that might otherwise complicate the interpretation of the results. If so, how stable are the RaRity estimates for RVs and CVs? At least the population can be stratified by sex to demonstrate the reproducibility (e.g. height).
2. How does RaRity perform with respect to sample size? Is there a minimum sample size required? I'd like to see some simulation results with respect to varying sample size, allele frequency thresholds, number of variants/genes, and underlying trait heritabilities.
3. The choice of 5000 RV block seems to be arbitrary. I guess that this number will be sensitive to the number of total individuals being used. How many genes will 5000 adjacent RV capture? For hox gene clusters, histone gene clusters, protocadherin gene cluster, TCR gene clusters, hemoglobin gene clusters and olfactory gene clusters, do they show different patterns? Do genes that are encoded from opposite orientation differ?
4. The authors say that RaRity has no MAF restriction, so I'd like to see how it performs for common variants with respect to LDSC and GCTA.
5. The authors define rare variants as having MAF <1%. I would want to see how different the method performs at different MAF thresholds. For example, if the estimates are based on 1) only RVs <0.001 or 2) RVs between 0.001 and 0.01, are they comparable?
6. For calibrating RaRity, can we expect a) the parameters chosen to reflect actual gene regions, and b) all gene regions to be consistent? It would be good to show some results in this section that show this.
7. Typically, start-gain, stop-loss, and splicing are not counted as strict LoF, as all these three categories are less pathogenic than nonsense, and frameshift mutations (e.g., refer to ClinVar). Please consider dropping them from LoF categories. Please also comment why inframe indels are not evaluated in this study.
8. There are other ways to detect linked RVs (e.g. gnomAD MNPs) in addition to PLINK. I wonder if the authors can check if the distant RV in LD can be detected with the large UK Biobank data.
9. Please explain the low concordance for RV and CV based heritability estimate for Lipoprotein (a) in Supp. Figure 3.
10. I'd like to see some effort towards replication of the results.

Minor questions

1. Does this study include multi-allele SNPs or only biallelic SNPs?
2. Does Minor Allele Count (MAC) mean the number of supporting reads?
3. Does gene length from UCSC RefSeq mean the transcript length or CDS length?
4. The first paragraph of the introduction is very long. Consider breaking this into two for ease of reading.
5. Line 63 of the first paragraph of the introduction says that "A limitation of gene-burden tests is the assumption that all RVs influencing a trait are homogeneous in terms of direction and magnitude of

effects." This fails to account for other gene-based rare variant testing methods, particularly variance-component tests (notably SKAT) and combined tests (SKAT-O), which are perhaps more commonly used and do allow for multiple directions of effect. Table 2 of the cited paper describes both of these subclasses of rare variant methods. These should be discussed as well.

6. The method overview and calibration sections in the Results section is a bit technical and Methods-like and don't really belong there, unless results are being presented.

7. Consider using the same color theme for Figure 1B and Figure 2A, and throughout the manuscript for the three themes of RV heritability estimates

8. Suppl. Figure 2 legend: it should be "PCs=Principal Components".

9. For Supplementary Figure 3, which version of the RV heritability estimates (in Fig 2) was used to compare with CV based estimate?

10. Line 242 on page 11 says that "the representation of each set of genes with significant contribution to the heritability". Does this mean statistical significance? If yes, describe how this was determined and include p-values. If not, a different word should be used.

11. Line 280 in the discussion mentions SKAT but says "it is solely designed for association testing, rendering it incompatible with RARity". Presumably this means that SKAT only tests strength of association via p-values without providing effect size, but this should be explained more clearly. Additionally, "incompatible" is perhaps the wrong word – "comparable" is probably better.

12. Line 410: "mean imputation was employed to fill in the missing genotypes". For rare variants, is mean that imputation really sufficient? Why not an approach mimicking what was done for the genotype data used for the common variants? Also, why were the genotypes standardized to mean 0, variance 1?

Reviewer #2 (Remarks to the Author):

This paper proposes RARity, a method to estimate rare-coding variant heritability. RARity uses a block-based approach that calculated the adjusted R^2 of each block and combined them to calculate overall heritability. It is different from the existing approach of using a linear mixed model. This block-based approach is valid only when each block is independent, so the authors used LD-pruning. The authors applied the method to UK-Biobank data of 37 traits and showed that rare coding variants can explain a substantial amount of heritability. Estimating rare-variant heritability is important, but the authors need to show the advantage of the proposed approach over existing methods. Also, there are some questions in UKBB heritability estimation, as the estimated values are larger than the existing results.

1. Can the estimate from the analysis be interpreted as heritability due to rare-coding variants, as rare variants that do not exist in WES data are not included in the analysis? The estimated heritability may be partly due to rare regulatory variants, which are in LD with rare-coding variants.

2. Heritability estimation is commonly done with a mixed-effect model. The authors discussed some limitations of the existing approaches in the Introduction. Still, I think the authors should apply them in simulations and real data analysis to compare them with RARity.

3. Since the method relies on several assumptions, running more extensive simulations with the different contributions of rare-coding variants would be good to evaluate the proposed methods. Currently, the authors considered only one simulation setup ($h^2_{rv}=0.05$, 20% of variants being causal).

4. Table 1, h^2_{CV} was estimated from the block-based approach with less stringent R^2 . It may inflate the h^2 estimate as this parameter caused inflation in rare variants. For example, height h^2_{CV} is 0.71, much higher than a recent estimate (~ 0.5).

5. The current approach is only applicable to continuous traits. The authors mentioned that the

extension to binary traits is future work, but it would be helpful if the authors provided a bit more information on how it can be done.

6. It isn't clear whether the current approach can be applicable to WGS data. Some discussion would be helpful.

7. The authors mentioned that one of the benefits of gene-based heritability estimation is that it can be used to find gene-level associations. But extensive literature exists for gene-based rare variant tests, and the method used in RARity (F-test) is less developed. For example, it cannot account for sample relatedness. In addition, some of the existing methods were developed to address the limitation of the F-test, which can have low power due to testing too many variants (large degree of freedom of the test). So I don't think RARity would be a useful tool for gene discovery as there are better alternatives.

RESPONSE TO REVIEWERS' COMMENTS

In this documentation, the reviewers' comments are presented in black, response to the reviewers' comments are written in blue italics and direct excerpts from the updated manuscripts, reflecting changes from the previous submission, are presented in purple.

Reviewer #1 (Remarks to the Author):

Estimating heritability without pedigrees is challenging. The authors provide an interesting and useful new method called RaRity for estimating heritability with rare variants for complex traits in large cohorts. In addition to aggregated RV count per gene, they proposed two frameworks to use unaggregated RVs either by sliding window of 5000 RVs or variable windows of RVs defined by genes. They also compared the RV based heritability with common variants (CV) based heritability in the UK Biobank cohorts with dozens of complex traits. They also showed the utility of this new measurement in the novel gene discoveries. Overall, the method is described well, and the results are compelling. Here are some concerns and suggestions for consideration to improve the clarity of the paper.

We sincerely thank the reviewer for their support and appreciate the time they took to review the manuscript and provide feedback. The comments have helped us make significant changes that strengthened our work. We hope that our responses to the comments thoroughly addresses all concerns.

Major concerns

Comment 1.1: All studied individuals in this study are unrelated Caucasian participants. Please show the PCA plots to see if there are sub-structures of the populations that might otherwise complicate the interpretation of the results. If so, how stable are the RaRity estimates for RVs and CVs? At least the population can be stratified by sex to demonstrate the reproducibility (e.g. height).

Response to Comment 1.1: *We thank the reviewer for pointing out the importance of genetic principal components as covariates. Indeed, a previous study by Galinsky et al., 2016 on 113,851 UKB samples have extensively studied the population structure in UKB, which have now been cited in the manuscript. Galinsky, et. al have shown that the population structure in the UK is dominated by five principal components (PCs) spanning six geographic clusters: Northern Ireland, Scotland, northern England, southern England, and two Welsh clusters, which was calculated using state-of-the art methods and presented in form of PCA plots¹. In this study, variations inferred by age, sex and the first 20 genetic PCs are accounted for by residualizing the traits with these potential covariates instead of stratified analyses, so as not to lose power. Simulations conducted under different conditions for rare coding variants (RVs) and comparison with other methods for common variants (CVs) demonstrate that PCA informed substructures do not complicate the interpretation of results. This is because RaRity estimates remain stable even after adjusting for covariates, including PCs. These findings are discussed in response to Comment 1.2 (b) and Comment 1.4.*

Furthermore, we acknowledge that there is some evidence for the impact of sex on common variant heritability estimates²⁻⁶. However, the impact of these variations on h^2_{RV} has never been explored. Sex stratification, performed on all 31 traits, showed some heterogeneity in h^2_{RV} between the sexes (Supplementary Figure 4), but the apparent differences were statistically nonsignificant (p -value > 0.05). Given these new insights, further exploration of the effects of sex stratification with a larger sample size is warranted. This new result is now included in the manuscript.

Supplementary Fig.4: Comparison of RV exome-wide heritability estimates between the sexes for 31 continuous traits. Estimation of heritability based on RVs with MAF <0.01 in 92,963 females and 74,385 males. Red bars indicate the standard errors of the heritability estimates. Differences in heritability estimates between the sexes appear heterogenous but remains statistically non-significant (p -value > 0.05).

METHODS

Biomarker and anthropometric data

Updated main text page 20; lines 432-435: All traits were then quantile transformed to resemble a standard normal distribution and further adjusted for age, sex, and the first 20 genetic principal components (PCs) to account for any effects of sub-population structure within UKB⁵², and finally standardized to have mean= 0 and variance= 1. Biomarker treatment for sex-stratified analysis was performed in an identical manner.

RESULTS

RV contribution to heritability estimates for 31 complex traits

Updated main text page 8; lines 169-171: Sex stratification, performed on all 31 traits, showed some heterogeneity in h^2_{RV} between the sexes (Supplementary Figure 4), but the apparent differences were statistically nonsignificant (p -value > 0.05).

DISCUSSION

Updated main text page 17; lines 371-376: We did not find any significant difference in the RV heritability between males and females across the 31 quantitative traits studied (Supplementary Fig.4). In contrast, SNP-based studies showed slight differences in heritability of selected traits between the sexes⁴⁰⁻⁴². Since the analysis using RVs was limited by the reduced power, further research of larger sample sizes is required to better understand the sex effect on RV heritability.

Comment 1.2 (a): How does RARity perform with respect to sample size? Is there a minimum sample size required?

Response to Comment 1.2: *We appreciate this valuable question, as the determination of an ideal sample size dictates the power and precision of the estimates generated. We developed a simple method using noncentrality parameters of the F-test statistics to simulate 10,000 adjusted R^2 (see Methods section) and estimated the power of RARity based on varying sample sizes and the targeted h^2_{RV} (Supplementary Figure 2). Using this method, we found that the sample size used in this study had 80% power to detect $h^2_{RV} = 4\%$ at an alpha-level of 0.05.*

METHODS

Statistical power

Updated main text page 25; lines 546-552: Statistical power of h^2_{RV} was estimated empirically from the variance of 10,000 simulated h^2_{RV} ($\widehat{Var}(\widehat{h}^2)$), under 230 conditions of sample sizes and true set h^2_{RV} . Non-central F -distributions were used to simulate the observed genetic effects at each genotype block, and exome-wide h^2_{RV} was derived as previously described. The true set h^2_{RV} ranged from 2% to 25%, with increments of 5%. Sample size was varied from 25,000 to 250,000 individuals by increments of 10,000. For each condition, the statistical power was calculated as the proportion of observed p -values less than 0.05 out of the 10,000 simulations.

RESULTS

Overview and testing of the RARity method.

Updated main text page 6; lines 122-123: The current sample size provided at least 80% power to detect 4% h^2_{RV} at an empirical type-I error rate of 0.05 (Supplementary Fig.2).

Supplementary Fig.2: Statistical Power as a function of sample size for different levels of RV heritability estimates. The horizontal dashed line corresponds to an empirical 80% power with an alpha-level of 0.05 using 10,000 simulation runs. The vertical dashed line marks the sample size used in the current study.

Comment 1.2 (b): I'd like to see some simulation results with respect to varying sample size, allele frequency thresholds, number of variants/genes, and underlying trait heritability.

Response to Comment 1.2(b): *We thank the reviewer for this suggestion requesting an increase in the breadth of simulations. Please refer to the above comment 1.1 regarding RARity performance with respect to varying sample sizes. We have conducted additional simulations to demonstrate RARity robustness with varying allele frequency thresholds, number of variants/genes, and underlying trait heritability. Our results showed that the estimates remain largely unbiased in all conditions tested. Only when the fraction of causal variants/genes was 1% or lower did we see a lower h^2_{RV} than expected. Our results demonstrated that RARity can provide robust estimates for h^2_{RV} across a variety of conditions.*

The results are presented below with the appropriate excerpts from the main text:

Supplementary Fig.3: Simulation of rare variant heritability estimates under varying LD pruning, fraction of causal genes and variants, and MAF conditions. Calibration of RARity for linkage disequilibrium, where r^2 = ‘coefficient of correlation’ and w = ‘window size’ thresholds used for LD pruning, (a) assuming homogenous distribution of heritability across all exome-wide blocks, and (b) assuming 10% of causal variants in 5% genes. In both cases, unbiased estimation of h^2_{RV} was observed when RVs were pruned with $r^2 > 0.1$ within a window size of 50Mb (blue arrows) and is the default LD pruning threshold used for all analyses. Figures (c-f) examined the sensitivity of RARity to varying (c) true h^2_{RV} values; (d) fraction of causal genes with 10% RVs with effects; (e) fraction of causal RVs in 5% of the genes and (f) MAF thresholds assuming 10% causal RVs within 5% of genes has an effect. Each dot in the figure represents a single simulation, with 20 exome-wide simulations performed for each scenario. The red vertical lines represent the 95% CIs, and the blue dashed, horizontal lines represent the true h^2_{RV} . Except for (f), RVs with $MAF < 0.01$ were selected for all analyses.

METHODS

Calibrating RARity

Updated main text page 26; lines 575-583: We generated 20 phenotypes (Y_{sim}) under each of the 7 scenarios to assess the overall impact of LD pruning parameters on RARity estimates and proceeded to estimating exome-wide h^2_{RV} as previously described. Additionally, the calibration was repeated under the assumption that the true h^2_{RV} originated from 10% of RVs within 5% of the genes, instead of homogenous distribution of h^2_{RV} across all blocks.

To test whether the more relaxed LD pruning for CVs ($r^2 > 0.9$, $w_s=1\text{Mb}$, step size= 500 bases) would bias the estimation of h^2_{CV} , we benchmarked RARity against alternative methods designed for CVs, namely BOLT²² and LDSC²³.

Testing the effects of genetic architecture

Updated main text page 27; lines 587-598: To assess whether RARity is sensitive to MAF thresholds, values of true h^2_{RV} , or varying fractions of causal variants or genes, we further tested RARity with simulations utilizing real genotype data, pruned with the default LD pruning threshold ($r^2 > 0.1$ within a window size of 50Mb). For each simulation, exome-wide h^2_{RV} was estimated from the simulated phenotype (Y_{sim}). A total of 20 simulations were carried out for each scenario as follows. The effect of varying MAF was tested by assuming that 10% of variants within 5% of genes are causal in each of the MAF categories (MAF<0.001, <0.005 and <0.01). The effect of varying h^2_{RV} was tested for MAF < 0.01 with the true h^2_{RV} set to either 2, 5, 10, 20 or 25 percent. Next, keeping a consistent MAF < 0.01 and true h^2_{RV} at 10%, the effect of varying fractions of causal genes (0.01, 0.05, 0.1 and 0.2) was tested with 10% of RVs with true effects. Finally, keeping a consistent MAF < 0.01, and true set h^2_{RV} at 10%, we tested the effect of varying fractions of causal variants (0.001, 0.05, 0.1 and 0.2) in 5% of randomly selected genes.

RESULTS

Overview and testing of the RARity method

Updated main text page 6; lines 123-133: Extensive simulations were performed using real genotype data to identify an approach for estimating heritability that is robust under realistic scenarios. Through this endeavour, we discovered that h^2_{RV} could be affected by the presence of long-range LD (LRLD), occurring at a much greater distance than what is observed for CVs²⁰ (Supplementary Fig. 3). LRLD complications are controlled by using a stringent LD threshold over a large, empirically derived window size ($r^2 > 0.1$, window size = 50Mb, step size = 500 base). This stringent pruning method was enforced in all subsequent analyses involving RVs to ensure a well-calibrated estimate of h^2_{RV} . The simulation studies on the effects of varying MAFs, heritability, proportion of causal genes or variants indicated that RARity is largely unbiased but tends to underestimate when the number of causal variants or genes is low (<1%; Supplementary Fig. 3).

DISCUSSION

Updated main text pages 12-13; lines 269-272: RARity is a versatile method, as it does not make any prior assumptions about the genetic architecture of the selected variants, making it applicable to both common and rare variants. Calibration of RV contributions was empirically confirmed using extensive simulation studies and proved to produce robust estimates across a wide range of analytical scenarios.

Comment 1.3 (a): The choice of 5000 RV block seems to be arbitrary. I guess that this number will be sensitive to the number of total individuals being used. How many genes will 5000 adjacent RV capture?

Response to Comment 1.3 (a): *The reviewer is right that other block sizes may be used. While the choice of 5000 RVs was indeed arbitrary, it was a decision balancing computational complexity, and potential biases. The computational complexity of OLS estimators is $O(nm^2)$, which increases quadratically with the number of RVs in a block, and thus should be reasonably small to run for all blocks. To ensure unbiased results, we tested the impact of the block size on the overall h^2_{RV} using a subset of variants from chromosome 22. More specifically, the first 4 blocks of ~5,000 protein altering and LoF RVs ($MAF < 0.01$) in chromosome 22 were combined to create blocks of ~10,000 RVs and ~20,000 RVs. Across all 33 traits, we observed consistent results (Supplementary Fig 11) regardless of the block size, however, blocks of 5,000 variants offered better computational speed. In conclusion, 5,000 variants per block was ultimately chosen for better computational efficiency, while maintaining accuracy.*

The blocks of 5000 RVs are gene-agnostic, i.e., not dependent on gene-borders, therefore the number of genes in each block is expected to vary. For example, the number of genes per exome-wide block with protein altering and LoF variants ($MAF < 0.01$) ranged from 17-170 genes. This information is now included in the manuscript, Methods.

Supplementary Fig.11: Impact of block size on RV heritability estimates. Estimation of heritability based on protein altering and LoF RVs with $MAF < 0.01$, using 20,000 variants from chromosome 22, partitioned into blocks of either 5,000, 10,000 or 20,000 consecutive variants. Red bars indicate 95% confidence intervals of the heritability estimates.

METHODS

Application to UKB data

Updated main text page 28; lines 611-625: 3) exome-wide blocks, were created by partitioning RVs in each chromosome into blocks of ~5000 adjacent RVs. This type of construct results in blocks that are gene-agnostic, i.e., independent of gene borders with varying number of genes per block, for example, the number of genes ranged from 17-170 per exome-wide block when protein altering and LoF variants (<0.01 MAF) were selected. The total heritability estimates from exome-wide blocks are denoted with h^2_{RV} . In each case, LD spillage was minimized through pruning prior to creating the blocks (as described above, in the “Genotype data quality control” section).

The impact of block size on h^2_{RV} was tested using the first 4 blocks of ~5,000 protein altering and LoF RVs (MAF <0.01) from chromosome 22, where 2 consecutive blocks were combined to create blocks of ~10,000 RVs and all four blocks were combined to form a single block of ~20,000 RVs. Since the size of blocks did not impact the accuracy of h^2_{RV} (Supplementary Fig 11), the choice of 5,000 RVs per block in an exome-wide block construct was motivated by computational efficiency.

Comment 1.3 (b): For hox gene clusters, histone gene clusters, protocadherin gene cluster, TCR gene clusters, hemoglobin gene clusters and olfactory gene clusters, do they show different patterns?

Response to Comment 1.3 (b): *Prompted by the reviewer’s comment, we estimated the h^2_{RV} for height in these clusters, but did not find an enrichment of h^2_{RV} contribution, nor did we observe any notable patterns. These results are presented in Supplementary Table 12.*

Supplementary Table 12: Heritability of height originating from RVs in selected gene clusters

Cluster	N GENES	N variants	RV heritability	LCL	UCL
Hemoglobin	10	159	0.000160256	-0.000159	0.00047947
Histone	77	2099	0.000414822	-0.0006753	0.00150498
HOX	226	14735	0.00117935	-0.0016787	0.00403741
Olfactory	366	18366	0.003147828	-6.97E-05	0.00636532
Protocadherin	59	7034	-0.00124205	-0.0031746	0.00069047

METHODS

Heritability estimates in relation to sex and gene-level annotations

Updated main text page 31; lines 683-687: To examine the enrichment of h^2_{RV} in highly conserved regions, we estimated the $h^2_{RV-gene-tot}$ of height originating from the conserved gene clusters, such as the hox, histone, protocadherin, and the hemoglobin gene clusters. The list of genes in these clusters was obtained from the HUGO Gene Nomenclature Committee (HGCN) database (www.genenames.org) and cross-listed with the genes carrying RVs in the UKB.

RESULTS

Characterizing genes based on heritability estimates

Updated main text page 12; lines 259-261: The highly conserved gene cluster regions, with short repeats, such as the *hox*, *histone*, *protocadherin* and *hemoglobin* gene clusters did not contribute significantly to h^2_{RV} (Supplementary Table 12)

Comment 1.3 (c): Do genes that are encoded from opposite orientation differ?

Response to Comment 1.3 (c): *Thanks to the reviewer's comment, we compared the $h^2_{RV-gene-total}$ contribution of the genes transcribed from the positive vs negative strand. The annotations for the gene directionality were obtained from the UCSC RegGene database¹⁰. The results show that $h^2_{RV-gene-total}$ of the genes encoded from opposite orientation are nearly identical (Supplementary Fig. 10). This is not surprising considering that there are roughly equal number of genes transcribed from either strand.*

Supplementary Fig.10: Comparison of RV heritability estimates between the genes encoded in the positive vs. negative strands. Estimation of heritability was based on RVs with MAF <0.01, using the gene-block. Red bars indicate the 95% confidence intervals of the heritability estimates.

METHODS

Heritability estimates in relation to sex and gene-level annotations

Updated main text page 31; lines 687-690: The impact of gene orientation on h^2_{RV} was examined by comparing the $h^2_{RV-gene-tot}$ contribution of the genes transcribed from the positive vs. negative strand, where the strand of each gene was annotated using the RefGene database in the UCSC genome browser⁵⁶, with roughly equal number of genes present on either strand.

RESULTS

Relationship with gene-length, gene-orientation and evolutionary constraint

Updated main text page 12; lines 261-263: The overall heritability estimates, $h^2_{RV-gene-tot}$, were consistent between genes that are transcribed from either the positive or the negative strand (Supplementary Fig. 10).

Comment 1.4: The authors say that RARity has no MAF restriction, so I'd like to see how it performs for common variants with respect to LDSC and GCTA.

Response to Comment 1.4: *We thank the reviewer for the suggestion of comparing RARity for h^2_{CV} with existing methods. We applied LDSC to estimate h^2_{CV} , as per the reviewer's request, but substituted GCTA¹¹ with BOLT-REML¹². The decision to substitute was primarily due to the better computational performance of BOLT-REML. Like the GCTA software, the BOLT-REML algorithm estimates heritability explained by common variants from partitioned genotyped SNPs and multiple traits measured among the same set of unrelated individuals and applies a variance component analysis to accomplish this. Unlike GCTA, BOLT-REML applies a Monte Carlo algorithm that is much faster and utilizes less computational memory at larger sample sizes, as opposed to the eigen decomposition-based methods used in GCTA¹², making BOLT-REML more efficient and feasible with our dataset.*

The results showed that RARity h^2_{CV} estimates were similar to BOLT-REML estimates, but generally higher than estimates derived using the LDSC method (Supplementary Fig. 5). While LDSC has the advantage of using summary level statistics and better control for population stratification, a number of papers have shown its limitations in underestimating h^2_{CV} ¹³⁻¹⁶.

Supplementary Fig.5: Comparison of methods to estimate common variant (MAF>0.01) heritability. The methods under comparison are RARity, LDSC, and BOLT in 31 traits that were adjusted for age, sex, and the first 20 PCs. See Methods for a description of each method. The red error bar represents 95% confidence intervals.

METHODS

Updated main text page 26; lines 581-583: To test whether the more relaxed LD pruning for CVs ($r^2 > 0.9$, $ws=1\text{Mb}$, step size= 500 bases) would impact the estimation of h^2_{CV} , we benchmarked RARity against alternative methods designed for CVs, namely BOLT²² and LDSC²³.

RESULTS

RV contribution to heritability estimates for 31 complex traits

Updated main text page 8; lines 173-176: Since RARity poses no upper restriction on the MAFs, we additionally assessed the contribution CV (h^2_{CV}) and the combined CV and RV (h^2_{tot}) to these 31 traits by concatenating common and rare variants (Methods). We observed that h^2_{CV} estimated using RARity was consistent with BOLT²², but higher than LDSC²³ (Supplementary Fig.5).

DISCUSSION

Updated main text page 13; lines 274-277: The performance of RARity for CV contributions was comparable with BOLT. The estimated h^2_{CV} by LDSC on the other hand was lower than RARity for each trait. While LDSC is a powerful tool that allows estimation h^2_{CV} from GWAS summary-statistics, with adequate control for population stratification, it is prone to underestimation as discussed in several studies

Comment 1.5: The authors define rare variants as having MAF <1%. I would want to see how different the method performs at different MAF thresholds. For example, if the estimates are based on 1) only RVs<0.001 or 2) RVs between 0.001 and 0.01, are they comparable?

Response to Comment 1.5: *As per the reviewer’s suggestion, we examined the contribution of different MAF categories to h^2_{RV} . The results show that MAF threshold <0.001 accounts for far more h^2_{RV} as compared to other MAF bins (0.01-0.005, 0.005-0.001 or 0.01-0.001). However, the per variant contribution, h^2_{RV} / RV , is the lowest for MAF <0.001, as this category also has the largest number of non-contributing variants.*

Supplementary Table 5: RV heritability for MAF bins calculated using gene-wise blocks

MAF	0.01 to 0.005				0.005 to 0.001				0.01 to 0.001				<0.001			
	N variants	$h^2_{RV-gene-tot}$	LCL	UCL	N variants	$h^2_{RV-gene-tot}$	LCL	UCL	N variants	$h^2_{RV-gene-tot}$	LCL	UCL	N variants	$h^2_{RV-gene-tot}$	LCL	UCL
ALB	6509	0.0063	0.0043	0.0084	23714	0.0093	0.0056	0.0131	29552	0.0146	0.0104	0.0188	1514538	0.0355	0.0066	0.0643
ALP	6509	0.0096	0.0075	0.0117	23714	0.0247	0.0208	0.0286	29552	0.0332	0.0288	0.0376	1514538	0.1110	0.0820	0.1400
ALT	6509	0.0040	0.0021	0.0060	23714	0.0136	0.0099	0.0174	29552	0.0177	0.0135	0.0220	1514538	0.0539	0.0250	0.0827
APOA1	6509	0.0049	0.0029	0.0069	23714	0.0143	0.0105	0.0181	29552	0.0181	0.0138	0.0223	1514538	0.0327	0.0038	0.0615
APOB	6509	0.0139	0.0117	0.0161	23714	0.0180	0.0142	0.0218	29552	0.0299	0.0256	0.0343	1514538	0.0710	0.0421	0.0999
AST	6509	0.0060	0.0040	0.0080	23714	0.0093	0.0055	0.0130	29552	0.0146	0.0104	0.0188	1514538	0.0398	0.0109	0.0686
CALC	6509	0.0040	0.0020	0.0060	23714	0.0077	0.0040	0.0114	29552	0.0120	0.0079	0.0162	1514538	0.0367	0.0079	0.0656
CHOL	6509	0.0101	0.0080	0.0122	23714	0.0146	0.0108	0.0184	29552	0.0232	0.0189	0.0275	1514538	0.0541	0.0253	0.0830
CREA	6509	0.0067	0.0047	0.0088	23714	0.0136	0.0098	0.0173	29552	0.0195	0.0152	0.0237	1514538	0.0450	0.0162	0.0739
CRP	6509	0.0062	0.0041	0.0082	23714	0.0134	0.0096	0.0171	29552	0.0188	0.0145	0.0230	1514538	0.0542	0.0253	0.0831
CYSC	6509	0.0064	0.0043	0.0084	23714	0.0128	0.0091	0.0166	29552	0.0177	0.0135	0.0219	1514538	0.0620	0.0331	0.0909
DBIL	6509	0.0084	0.0063	0.0105	23714	0.0094	0.0056	0.0131	29552	0.0171	0.0129	0.0213	1514538	0.0766	0.0476	0.1055
GGT	6509	0.0093	0.0072	0.0114	23714	0.0124	0.0086	0.0161	29552	0.0207	0.0165	0.0250	1514538	0.0491	0.0203	0.0780
GLU	6509	0.0032	0.0013	0.0052	23714	0.0046	0.0010	0.0083	29552	0.0079	0.0038	0.0120	1514538	0.0138	-0.0151	0.0426
HBA1C	6509	0.0099	0.0078	0.0121	23714	0.0164	0.0126	0.0202	29552	0.0258	0.0215	0.0301	1514538	0.0772	0.0483	0.1061
HDL	6509	0.0048	0.0028	0.0068	23714	0.0146	0.0108	0.0184	29552	0.0185	0.0143	0.0227	1514538	0.0515	0.0226	0.0804
IGF1	6509	0.0071	0.0050	0.0091	23714	0.0161	0.0123	0.0199	29552	0.0222	0.0179	0.0264	1514538	0.0542	0.0253	0.0831
LDL	6509	0.0113	0.0091	0.0134	23714	0.0151	0.0113	0.0189	29552	0.0245	0.0202	0.0288	1514538	0.0644	0.0355	0.0933
LPa	6509	0.0066	0.0045	0.0086	23714	0.0177	0.0138	0.0215	29552	0.0230	0.0187	0.0272	1514538	0.0235	-0.0053	0.0524
PHOS	6509	0.0078	0.0057	0.0099	23714	0.0113	0.0076	0.0151	29552	0.0180	0.0138	0.0223	1514538	0.0618	0.0329	0.0907
TBIL	6509	0.0095	0.0074	0.0116	23714	0.0112	0.0074	0.0149	29552	0.0202	0.0159	0.0244	1514538	0.0686	0.0397	0.0975
TP	6509	0.0091	0.0070	0.0112	23714	0.0099	0.0061	0.0136	29552	0.0179	0.0137	0.0221	1514538	0.0503	0.0214	0.0791
TRIG	6509	0.0073	0.0052	0.0094	23714	0.0116	0.0078	0.0153	29552	0.0182	0.0140	0.0224	1514538	0.0629	0.0340	0.0918
UREA	6509	0.0045	0.0025	0.0065	23714	0.0042	0.0005	0.0078	29552	0.0086	0.0045	0.0128	1514538	0.0197	-0.0091	0.0486
UA	6509	0.0059	0.0039	0.0079	23714	0.0099	0.0061	0.0136	29552	0.0151	0.0110	0.0193	1514538	0.0490	0.0202	0.0779
VITD	6509	0.0018	-0.0002	0.0037	23714	0.0066	0.0029	0.0103	29552	0.0083	0.0042	0.0125	1514538	0.0120	-0.0168	0.0409
SBP	6509	0.0053	0.0033	0.0073	23714	0.0089	0.0052	0.0126	29552	0.0135	0.0093	0.0177	1514538	0.0549	0.0260	0.0838
DBP	6509	0.0060	0.0040	0.0080	23714	0.0082	0.0045	0.0119	29552	0.0136	0.0094	0.0178	1514538	0.0783	0.0494	0.1072
BMI	6509	0.0054	0.0034	0.0074	23714	0.0089	0.0052	0.0127	29552	0.0141	0.0099	0.0183	1514538	0.0869	0.0580	0.1158
WTH	6509	0.0051	0.0030	0.0071	23714	0.0073	0.0036	0.0110	29552	0.0120	0.0079	0.0162	1514538	0.0647	0.0358	0.0936
HEIGHT	6509	0.0171	0.0148	0.0193	23714	0.0301	0.0262	0.0341	29552	0.0460	0.0415	0.0506	1514538	0.1726	0.1435	0.2016

METHODS

Genotype data quality control

Rare coding variants:

Updated main text page 20; lines 452-453: RVs were also subset into MAF bins (MAF= 0.01-0.005, 0.005-0.001 or 0.01-0.001) to examine the contribution of different MAF categories to h^2_{RV} .

RESULTS

RV heritability to characterize pathogenicity scores

Updated main text page 10; lines 211-217: Further, the allele frequency of variants seemed to have very little impact on the magnitude of enrichment, as we observed similar results across the three MAF categories (< 0.001, < 0.005, < 0.01; Fig. 4, Supplementary Fig. 7). The specified MAF cut-offs were used instead of MAF-bins as the $h^2_{RV-gene-tot}$ in these bins ($0.01 > \text{MAF} \geq 0.005$, $0.005 > \text{MAF} \geq 0.001$, and $0.01 > \text{MAF} \geq 0.001$) were <5% (Supplementary Table 5) and consequently would have produced unreliable results when further stratified by pathogenicity scores.

Comment 1.6: For calibrating RARity, can we expect a) the parameters chosen to reflect actual gene regions, and b) all gene regions to be consistent? It would be good to show some results in this section that show this.

Response to Comment 1.6: *For calibration of RARity, gene regions do not have a noticeable effect. During the initial submission, we assumed a homogenous distribution of h^2_{RV} across blocks of 5000 variants, without making further assumptions about the genetic regions. Since the last submission, we have tested 23 additional scenarios, with 20 simulations per scenario, including the selection of various fractions of causal genes (Supplementary Fig. 3 E). The results of these scenarios are now discussed in the manuscript and under Response to Comment 1.2(b).*

Comment 1.7: Typically, start-gain, stop-loss, and splicing are not counted as strict LoF, as all these three categories are less pathogenic than nonsense, and frameshift mutations (e.g., refer to clinvar). Please consider dropping them from LoF categories. Please also comment why inframe indels are not evaluated in this study.

Response to Comment 1.7: *A careful review of the sentences pertaining to the inclusion criteria of variants revealed a few ambiguities in the referred sentence. We have indeed included in-frame INDELS in the study but failed to mention it in the initial manuscript. Additionally, start-gain and splicing variants were not included in the LoF category, but stop-loss were. We deeply apologize for this lack of clarity and have now made the correction in the following manner to improve clarity:*

METHODS

Genotype data quality control

Rare coding variants:

Updated main text page 20; lines 446-452: Qualifying RVs were defined as variants that were nonsynonymous single nucleotide variants, frameshift deletions or insertions, in-frame deletions or insertions, stop-gain, stop-loss and start-loss variants, with a minor allele count (MAC) >2 and MAF below the cut-off (< 1%, <0.5% or <0.1%) in all gnomAD subpopulations, and locally in the UKB samples. Within these variants, the stop gain/loss variants and frameshift variants were defined as the LoF variants, and the rest are referred to as protein-altering variants.

Comment 1.8: There are other ways to detect linked RVs (e.g. gnomAD MNPs) in addition to PLINK. I wonder if the authors can check if the distant RV in LD can be detected with the large UK Biobank data.

Response to Comment 1.8: *GnomAD multi-nucleotide polymorphism (MNPs), which is now widely known as gnomAD multi-nucleotide variants (MNVs) are identified by searching for variants that appear in the same individual, in cis, and within 2 bp distance for the exome data set and 10 bp distance for the genome data set, checking every pair of genotypes within a certain window size, for every individual, to identify whether the individual carries pair(s) of mutation in the same haplotype¹⁷. This is a state-of-the-art method to identify short-range variants existing on the same haplotype in the same individual, but it will not solve the issue of long-range LD (at a distance of 50Mb) that interferes with the estimation of RV heritability. LD pruning using PLINK, was done using the Pearson's correlation (linkage disequilibrium) between pairwise variants, over a 50MB window size, retaining variants with $r^2 < 0.1$, making it a more stringent and robust approach to handle long-range LDs that are not usually detected by other methods. Furthermore, since both LD pruning using PLINK and RARity utilize the correlation structure, LD pruning using PLINK is more suitable for RARity, as compared to other methods of detecting LD between variants.*

Comment 1.9: Please explain the low concordance for RV and CV based heritability estimate for Lipoprotein (a) in Supp. Figure 3.

Response to Comment 1.9: *Supplementary Fig.3 in the initial submission, now referred to Supplementary Fig.6, shows that although common variants contributed more to overall heritability, as compared to RVs, h^2_{RV} generally increased with h^2_{CV} , except for height, alkaline phosphatase and Lp(a). Lp(a) particularly stands out with a much higher h^2_{CV} in relation to h^2_{RV} . The low concordance between CV and RV heritability in Lp(a) may be due to the unique genetic architecture of this trait. The LPA gene (encoding for lipoprotein(a)) largely determines the concentrations of the Lp(a) plasma protein, but the presence of highly polymorphic copy number variations of the kringle IV repeats also plays a role¹⁸. This is now addressed in the manuscript.*

RESULTS

RV contribution to heritability estimates for 31 complex traits

Updated main text pages 8-9; lines 176-182: Although common variants contributed more to overall heritability, as compared to RVs (Table 1), estimated h^2_{RV} generally increased proportionally to h^2_{CV} (Supplementary Fig.6), except for height, alkaline phosphatase and Lp(a). Lp(a) particularly stands out with a much higher h^2_{CV} in relation to h^2_{RV} . The low concordance between CV and RV heritability in Lp(a) may be due to the unique genetic architecture of this trait, with most genetic variance attributed to the LPA locus itself and the highly polymorphic kringle IV type 2 copy number variation having an outsized impact on concentration²⁴.

Comment 1.10: I'd like to see some effort towards replication of the results.

Response to Comment 1.10: *We thank the reviewer for this important consideration. We have extensively replicated the simulation studies by using 23 scenarios with 20 simulations or replicates for each (Response to Comment 1.2(b)). Each simulated outcome per scenario was generated by using the real UKB exome genotypes to capture real-world genetic architecture. While replication of the results in other biobanks with extensive exomes and phenotyping would be invaluable, we regret to inform that we currently do not have access to any other compatible or comparable dataset (in terms of ethnicity, sample size, and measurement of traits) that would enable us to confirm the findings of this study. This is an important question, and we hope that as newer biobanks become publicly accessible topics within this study could be further explored.*

Minor questions

Comment 1.11: Does this study include multi-allele SNPs or only biallelic SNPs?

Response to Comment 1.1: *Yes, this study includes multi-allele variants, however, multi-allelic variants were treated as bi-allelic by considering the presence or absence of (any) rare variant. This information is now included in the manuscript.*

METHODS

Genotype data quality control

Rare coding variants:

Updated main text page 21; lines 462-465: Individual level genotypes were extracted with PLINK1.9, assuming an additive model for all variants, and thus allotting a score of 2 for rare allele homozygous variants, 1 for heterozygous variants, and 0 otherwise. Multi-allelic variants were treated as bi-allelic by considering the presence or absence of (any) rare variant.

Comment 1.12: Does Minor Allele Count (MAC) mean the number of supporting reads?

Response to Comment 1.12: *We apologize for not providing a clear definition of the minor allele count (MAC) in the original manuscript, therefore, we have now explicitly defined MAC in the manuscript, as the total number of minor alleles at each locus in the population being studied.*

METHODS

Genotype data quality control

Rare coding variants:

Updated main text page 20; lines 446-452: Qualifying RVs were defined as variants that were nonsynonymous single nucleotide variants, frameshift deletions or insertions, in-frame deletions or insertions, stop-gain, stop-loss and start-loss variants, with a minor allele count (MAC, the number of minor alleles at each locus in the population being studied) >2 and MAF below the cut-off (< 1%, <0.5% or <0.1%) in all gnomAD subpopulations, and locally in the UKB samples. Within these variants, the stop gain/loss variants and frameshift variants were defined as the LoF variants, and the rest are referred to as protein-altering variants.

Comment 1.13: Does gene length from UCSC RefSeq mean the transcript length or CDS length?

Response to Comment 1.13: *In the analysis where we look at the relationship between gene-level heritability and gene length, we used the length of the largest transcript as a proxy for the gene length. As mentioned in the manuscript:*

METHODS

Heritability estimates in relation to sex and gene-level annotations

Updated main text page 31; lines 677-679: For each gene, the transcript with the longest transcript length was selected to represent the length of the gene. Since the distribution of the gene lengths are skewed, with some genes being much longer than others, the lengths of all genes were log transformed.

Comment 1.14: The first paragraph of the introduction is very long. Consider breaking this into two for ease of reading.

Response to Comment 1.14: *We thank the reviewer for their suggestion. We have restructured the first paragraph into multiple sections for ease of reading.*

Comment 1.15: Line 63 of the first paragraph of the introduction says that “A limitation of gene-burden tests is the assumption that all RVs influencing a trait are homogeneous in terms of direction and magnitude of effects.” This fails to account for other gene-based rare variant testing methods, particularly variance-component tests (notably SKAT) and combined tests (SKAT-O), which are perhaps more commonly used and do allow for multiple directions of effect. Table 2 of the cited paper describes both of these subclasses of rare variant methods. These should be discussed as well.

Response to Comment 1.15: *We appreciate the reviewer’s suggestion and have discussed the differences between gene-burden tests, SKAT and SKAT-O in greater details in the introduction.*

INTRODUCTION

Updated main text page 3; lines 50-65: Genome-wide association studies (GWAS) have been fruitful for characterizing common variants with regards to complex traits; however, a similar approach lacks the statistical power to study rare variants, unless sample sizes or effect sizes are very large⁷. Consequently, to improve statistical power, RV association testing often relies on gene-level variant aggregation methods to perform gene-burden tests, or variance component tests such as Sequence Kernel Association Test (SKAT)⁸ and its variations (e.g., SKAT-O or “Optimal SKAT”). A limitation of gene-burden tests is the assumption that all RVs influencing a trait are homogeneous in terms of direction and magnitude of effects⁷. SKAT, on the other hand, aggregates the associations between variants and the phenotype through a kernel matrix⁸, is a powerful tool for association testing in the presence of variants acting in opposing directions, but can be less powerful than burden tests when most variants are causal, and effects are in the same direction. SKAT-O combines both burden and SKAT to overcome the limitations of SKAT but can be slightly less powerful than burden or variance-component tests if their assumptions are held. While SKAT and SKAT-O are solely designed for association testing, variant aggregation in gene-burden testing may be used in any scenario requiring a gene to be treated as a single unit. However, the impact of RV aggregation on phenotypic variance explained has never been empirically evaluated.

Comment 1.16: The method overview and calibration sections in the Results section is a bit technical and Methods-like and don’t really belong there, unless results are being presented.

Response to Comment 1.16: *We thank the reviewer for the helpful suggestion. We have removed most of the technical elements from the results section, leaving only the most relevant information to provide context and emphasis on the results being presented.*

Comment 1.17: Consider using the same color theme for Figure 1B and Figure 2A, and throughout the manuscript for the three themes of RV heritability estimates.

Response to Comment 1.17: *We thank the reviewer for this suggestion. A more consistent colour scheme has been used throughout the manuscript.*

Comment 1.18: Suppl. Figure 2 legend: it should be “PCs=Principal Components”.

Response to Comment 1.18: *We thank the reviewer for the correction, we have made the necessary change in this figure highlighted in the caption below (now referred to as Supplementary Fig.1).*

Supplementary Fig. 1: Summary of the Rare variant heritability (RARity) estimator pipeline. The RARity pipeline constitutes pre-treatment of the genotype and the phenotype data, followed by application of the statistical model for each block and finally estimation of the total heritability from all blocks. *The model may be modified to prioritize variants by implementing LD clumping instead of pruning. In addition, the pruning parameters are dependent on the selection of common vs rare variants for the analysis. **Adjustment of medications may involve implementing correction factors or removal of individuals using the medications. MAF=minor allele frequency, MAC=minor allele count, LD= linkage disequilibrium, SD=standard deviation, **PCs=principal components**.

Comment 1.19: For Supplementary Figure 3, which version of the RV heritability estimates (in Fig 2) was used to compare with CV based estimate?

Response to Comment 1.19: *Exome-wide blocks were used to estimate h^2_{RV} in Supplementary Figure 3 (which is now referred to as Supplementary Figure 6). We apologize for not making this clear in the figure. We have now made the correction.*

Supplementary Fig. 6: Correlation between the contribution of RV and CV to complex traits heritability. Both CV and RV contributions to heritability was estimated with RARity, however the LD pruning threshold was varied (Methods) depending on the variant type. Exome-wide blocks with 5,000 RVs/ block were used to estimate h^2_{RV} , while genome-wide blocks with 20,000 CVs/block were used to estimate h^2_{CV} . Shaded area indicates 95% CI.

Comment 1.20: Line 242 on page 11 says that “the representation of each set of genes with significant contribution to the heritability”. Does this mean statistical significance? If yes, describe how this was determined and include p-values. If not, a different word should be used.

Response to Comment 1.20: *We apologize for the lack of clarity in pathway analysis. We aimed to demonstrate the functional relevance of genes with significant h^2_{RV} (p -value < 2.75×10^{-6}), in biological pathways, diseases and pharmacogenomics. Realizing the ambiguity in this analysis, we performed the g:profiler analyses once more, this time using the web-based tool and strictly following the recommendations outlined in the g:Profiler website. To demonstrate the pharmacogenomic importance of the genes with significant RV heritability towards traits, we ran a query of the target genes in the Drug Genome Database Interaction database (DGIdb), and thus identifying the significant heritability genes belonging to the “druggable genome” and “clinically actionable” categories. Disease pathways were explored using DisGeNET. Please see the following modifications to the manuscript.*

METHODS

Updated main text page 29-30; lines 650-667:

Network and pathway analyses: g:GOST, the g:Profiler⁶¹ web tool for functional profiling, was used to test the enrichment of the genes with significant heritability ($h^2_{RV-gene} p\text{-value} < 2.75 \times 10^{-6}$), against gene-sets in common databases. The significant heritability genes for each trait were treated as separate gene lists for independent query, and statistical tests were conducted within a domain scope of only annotated genes, considering the GO biological process, GO molecular function, GO cellular component, KEGG, Reactome, TRANSFAC, miRNA, CORUM, HP, HPA, and WikiPathways data sources, and removing electronic GO annotations. This analysis resulted in a list of statistically significant enriched terms for each gene list, adjusted for multiple testing using g:SCS (set counts and sizes) $p\text{-value} < 0.05$, integral to the g:Profiler server. An Enrichment Map was then created with the results from g:Profiler, using Cytoscape version 3.10.1⁶², with an FDR q-value cut-off value set to 0.001, and medium connectivity, producing networks where the nodes represent enriched pathways, and the edges represent all pairwise connections. These threshold values were chosen to ensure that we captured highly enriched pathways while being able to observe structure in the network. Next, we clustered the nodes to distinguish the major theme in the enriched pathways for each trait. DisGeNET²⁵ was used to retrieve and explore the gene-disease associations for the significant list of genes, resulting in a disease-heat-map for each trait. Additionally, we ran a query of the target genes in the Drug-Gene Interaction database (DGIdb)²⁶ to identify potentially druggable and clinically actionable genes.

RESULTS

Characterizing genes based on heritability estimates:

Updated main text page 11; lines 236-248: We further investigated the role of the 152 genes contributing significantly to trait heritability ($p\text{-value} < 2.75 \times 10^{-6}$) in diseases and biological pathways. DisGenet²⁵ revealed 115 of the target genes influencing 2,137 disease pathways (Supplementary Table 7). A heatmap of the diseases associated with the significant genes for ApoB is presented in Fig. 6a. Interestingly, searching through the Drug Gene Interaction database (DGIdb)²⁶ revealed that 93 of the 152 of the target genes (i.e., 61%) belong in the “druggable genome” category, 16 of which are “clinically actionable”, including *APOB*, *FGFR3* and *LDLR* (Supplementary Table 8). The target genes also appear to be significantly overrepresented in many biologically relevant pathways (multiple-testing correction via the g:Profiler g:SCS algorithm, adjusted $p\text{-value} < 0.05$). For example, the genes contributing significantly towards h^2_{RV} of ApoB results in enrichment of 156 pathways, all of which are highly interconnected in an elaborate network and includes well-known pathways such as the LDL receptor binding pathways and pathways related to atherosclerosis (Fig. 6b, Supplementary Table 9).

DISCUSSION

Updated main text page 15-16; lines 328-346: We further demonstrate that the 152 genes with significant $h^2_{RV-gene}$ across 31 traits are enriched in various biological and disease pathways. For example, 11 significant genes contributing to heritability of ApoB are involved in pathways ranging from abnormal arterial stenosis to chylomicron, LDL and lipoprotein clearance, as well as hyperlipidemia and coronary artery diseases (Fig. 6, Supplementary Tables 8 and 9). Interestingly, previous association studies did not identify the *PPARA* gene as genome-wide significant, even though *PPARA* significantly contributes to the h^2_{RV} of ApoB and that both *PPARA* and *APOB* are involved in several lipid-related pathways (Supplementary Table 9), fatty liver disease, and dyslipidemias (Fig. 6), and identified as “druggable genome” in the DGIdb database (Supplementary Table 7). Notably, *PPARA* is also the target of the lipid-lowering drug class known as fibrates³⁶, making it a strong candidate for further pharmacogenomic studies. Indeed, most of the significant genes play important roles in disease etiology, as observed through DisGenet analyses. Further examples include the contribution of *TFAM* on alkaline phosphatase

heritability and its role in hepatocerebral mitochondrial DNA depletion syndrome, and the contribution of *TMEM43* on HbA1c heritability and macular degeneration. These results imply that genes involved in diseases are also likely to contribute significantly to biomarker heritability. However, whether these associations can be used as markers of pathogenic mutations or represent causal mediations through biomarker concentrations will require further investigations.

Comment 1.21: Line 280 in the discussion mentions SKAT but says “it is solely designed for association testing, rendering it incompatible with RARity”. Presumably this means that SKAT only tests strength of association via p-values without providing effect size, but this should be explained more clearly. Additionally, “incompatible” is perhaps the wrong word – “comparable” is probably better.

Response to Comment 1.21: *We thank the reviewer for the helpful comment, as it has enabled us to clarify the rationale for why SKAT cannot be used with RARity, in the same manner as we have done for variant aggregation. RARity calculates heritability estimates, whereas SKAT/ burden testing is meant for association testing. We believe both methods should complement each other to uncover important information about RVs. In the manuscript, we empirically evaluate the effect of variant aggregation (used in gene burden testing) on trait variance. As suggested by the reviewer, we have changed the wording “incompatible” to incomparable.*

DISCUSSION

Updated main text page 14; lines 303-310: Several methods have been used to increase the power of detecting gene-trait associations, the most popular methods being gene burden testing and Sequence Kernel Association Test (SKAT)⁸. Since SKAT aggregates the associations between variants and the phenotype through a kernel matrix⁸ it is solely designed to test the strength of association via p-values without providing an effect size, and thus rendering it incomparable to RARity. The variant aggregation method used in gene-burden testing do not have such a limitation, and consequently, we were able to examine the effect of aggregating RVs on trait variance, which on average resulted in a 79.3% loss on the estimated h^2_{RV} . These results suggest that burden tests may have limited ability to predict traits.

Updated main text page 17; lines 378-382: In addition, heritability describes the phenotypic variance at the population level, and thus cannot be generalized to a different population, and nor does it inform the prediction of phenotypic variations of an individual. Therefore, methods to estimate RV heritability, such as RARity, are not intended to replace RV association methods, but rather to complement.

Comment 1.22: Line 410: “mean imputation was employed to fill in the missing genotypes”. For rare variants, is mean that imputation sufficient? Why not an approach mimicking what was done for the genotype data used for the common variants? Also, why were the genotypes standardized to mean 0, variance 1?

Response to Comment 1.21: *While it is possible to impute rare variants from genotype data, RV heritability estimates utilizing genotype data are limited by the selection of rare genetic variation captured on the array, and algorithms to impute missing variants results in low accuracy for truly rare alleles (MAF<0.05)^{26,27}. Whole exome sequences are thus ideal for studying the impact of rare coding variants on trait heritability. The use of whole exome sequence is thus rationalized in the Introduction section.*

The word “imputation” is used in two different contexts in the manuscript. Imputation metric used for CVs refer to the initial quality control step, where CVs originating from the microarray data were filtered to retain variants with imputation quality score greater than 0.7, details of the imputation quality scores are provided in the UK Biobank website.

During the initial quality control steps, we removed missing genotypes present in more than 90% of samples (m=369,215). In the sentence “mean imputation was employed to fill in the missing genotypes”, refers to using the mean imputation to fill in the missing RVs in the remaining <10 % samples. We apologize for the confusion and have now clarified it further in the manuscript.

The reviewer is correct to indicate that standardization of the genotype data is not required for RARity, as it is possible to estimate heritability on non-standardized genotypes. However, scaling all genotype data to mean=0, standard deviation =1 allows direct comparison of regression coefficients (betas) for the impact on phenotype sample variance and simplifies some of the downstream calculations.

INTRODUCTION

Updated main text page 4; lines 70-79: The RV contribution to narrow-sense heritability, h^2_{RV} , defined as the proportion of phenotypic variance attributable to their additive genetic effects, has been estimated in several recent studies¹¹⁻¹⁷. However, these studies are limited by either the use of genotyping array data, small sample size, or models that make specific assumptions about the underlying genetic architecture. RV heritability estimates utilizing genotype data^{12,13,15,16} are limited by the selection of rare genetic variation captured on the array, and algorithms to impute missing variants resulted in low accuracy for truly rare alleles^{16,18}. As next-generation sequencing, particularly, whole exome sequencing (WES), are becoming a common place to accurately detect RVs^{11,14,17}, methods to evaluate RV heritability that do not rely on assumptions about the genomic architecture are needed.

METHODS

Genotype data quality control

Rare coding variants:

Updated main text page 20; lines 439-441: All monomorphic variants (m=83,700), variants with missing genotypes in more than 10% samples (m=369,215), and those deviating significantly from Hardy-Weinberg Equilibrium (p -value $< 5 \times 10^{-6}$; m=35,317) were removed.

Updated main text page 21; lines 465-467: While RVs with missing genotype in >10% samples were removed during the early QC steps, mean imputation was employed to fill in the missing genotypes in the remaining samples.

Common variants: *Updated main text page 21; lines 472-474: The imputed genotypes are based on the Genome Reference Consortium Human Build 37 (GRCH37), and further filtered to retain CVs with imputation quality score greater than 0.7, those with no significant deviation from Hardy-Weinberg equilibrium (p -value $> 1 \times 10^{-10}$)*

Updated main text page 22; lines 479-482: Similar to RVs, individual level genotypes were extracted with PLINK1.9, assuming an additive model, mean imputation was also employed to fill in the missing genotypes (in <10% samples), followed by standardized to have mean= 0 and variance= 1.

Reviewer #2 (Remarks to the Author):

This paper proposes RARity, a method to estimate rare-coding variant heritability. RARity uses a block-based approach that calculated the adjusted R^2 of each block and combined them to calculate overall heritability. It is different from the existing approach of using a linear mixed model. This block-based approach is valid only when each block is independent, so the authors used LD-pruning. The authors applied the method to UK-Biobank data of 37 traits and showed that rare coding variants can explain a substantial amount of heritability. Estimating rare-variant heritability is important, but the authors need to show the advantage of the proposed approach over existing methods. Also, there are some questions in UKBB heritability estimation, as the estimated values are larger than the *existing results*.

We highly appreciate the reviewer's feedback and appreciation of the RARity method. We agree with the reviewer's comments and have provided detailed responses below.

Comment 2.1. Can the estimate from the analysis be interpreted as heritability due to rare-coding variants, as rare variants that do not exist in WES data are not included in the analysis? The estimated heritability may be partly due to rare regulatory variants, which are in LD with rare-coding variants.

Response to comment 2.1: *It is quite possible that some of the rare coding variants are in LD with the non-coding variants that are not represented in WES. However, numerous studies have shown that rare coding variants have major functional impacts in a direct way³⁴. On the other hand, very little is known about the rare non-coding regions, which are difficult to assess functionally³⁵, consequently there is no direct comparison of the contribution of rare-coding vs. non-coding variants in the literature. This is indeed a question of high importance and may be answered with WGS using RARity in the future. Thanks to the reviewer's comment, this is now explained as a limitation in the discussion section of the manuscript.*

DISCUSSION

Updated main text pages 16-17; lines 355-367: This study has several limitations. First, exclusion of non-coding, singleton, and doubleton variants, as well as LD pruning of potentially functional variants, may lead to an underestimate of the h^2_{RV} . Although, numerous studies have shown that rare coding variants have major functional impacts in direct way³⁷, it is quite possible that some of the observed h^2_{RV} is due to the coding RVs being in LD with the non-coding variants that are not represented in WES. On the other hand, very little is known about the rare non-coding regions, which are difficult to define and even more difficult to assess functionality³⁸, consequently there is no direct comparison of the contribution of rare-coding vs non-coding variants in the literature. This is indeed a question of high importance and may be answered with WGS. We anticipate that RARity can be applied to WGS with further calibrations, to study effects of other variants not discussed here. One of the challenges of WGS data analysis is that there is no natural biological unit available in the intergenic regions. With RARity, there is no need for defining the biological units, as the estimates can be based on blocks that are agnostic of genetic borders.

Comment 2.2. Heritability estimation is commonly done with a mixed-effect model. The authors discussed some limitations of the existing approaches in the Introduction. Still, I think the authors should apply them in simulations and real data analysis to compare them with RARity.

Response to comment 2.2: *We appreciate the feedback from the reviewer. In accordance with the request, we compare RARity with LDSC and BOLT-REML using real genotype data including common variants (Please see **Response to Comment 1.4**). Since other methods are geared towards CVs, rather*

than RVs, we did not test the methods through simulations, as it is outside the scope of this manuscript. Indeed, LDSC has not been developed for rare variants analysis and the use of BOLT-REML would be computationally challenging for the number of variants used in this study. Neither method has been validated for rare variants analysis.

Comment 2.3. Since the method relies on several assumptions, running more extensive simulations with the different contributions of rare-coding variants would be good to evaluate the proposed methods. Currently, the authors considered only one simulation setup ($h^2_{rv}=0.05$, 20% of variants being causal).

Response to comment 2.3: *We agree with the reviewer's perspective on the requirement of simulations to strengthen the evidence for the robustness of the model. As stated in the manuscript, RARity requires no parametrization nor assumptions regarding the genetic architecture of traits analyzed (such as polygenicity of effects or relationships between MAF/LD and effect size), but relies on the assumption of block-wise independence, which was addressed by a stringent LD pruning to avoid long-range LD. During the initial submission, we performed simulations for 7 different scenarios, where we assumed a homogenous distribution of h^2_{RV} across blocks of 5000 variants, without making any assumption about the genetic region, but varied the conditions of LD pruning. Since the last submission, we have tested 23 additional scenarios, with 20 simulations per scenario, including selection of various fractions of causal genes/variants, heritability estimates and MAFs (Supplementary Fig. 3 E). The results of these scenarios are now discussed in the manuscript and under **Response to Comment 1.2(b)**. We believe that implementing this suggestion has greatly strengthened the evidence towards RARity being a robust method for estimating RV heritability.*

Comment 2.4. Table 1, h^2_{CV} was estimated from the block-based approach with less stringent R^2 . It may inflate the h^2 estimate as this parameter caused inflation in rare variants. For example, height h^2_{CV} is 0.71, much higher than a recent estimate (~ 0.5).

Response to Comment 2.4. *The reviewer makes an important point. Unlike RVs, CVs do not require a stringent LD pruning threshold, as they are not affected by very long-range LD. We came to this conclusion through extensive simulations, presented in another manuscript on gene-by-environment interactions, using a similar framework and LD pruning threshold for CVs. This manuscript is currently published in Nature Communications (<https://www.nature.com/articles/s41467-023-40913-7>)¹⁶.*

It is quite challenging to compare heritability estimates based on the literature values as the methods and samples differ between populations. Srivastava, et. al has recently compared the h^2_{CV} for height and BMI using GREML, LADK, LDSC and SumHer and their variations, while utilizing individual-level data from The Northern Finland Birth Cohort (NFBC) and summary results from the UK Biobank and Genetic Investigation of Anthropometric Traits (GIANT) consortium. Their results showed that the estimates for h^2_{CV} of height ranged from 0.4552 - 0.6785 for height, and 0.1908 - 0.2844 for BMI, depending on the method used¹⁵. A similar comparison of GRE, LDSC, S-LDSC and SumHer was conducted by Hou, et. al, utilizing 290K unrelated British individuals from the UK Biobank, where they compared the h^2_{CV} of 22 complex traits, including height and BMI. Their results showed the estimates of h^2_{CV} of height ranged between 0.55-0.73, and that of BMI ranged between 0.285-0.436³⁶. Yet another study using UK Biobank showed that the h^2_{CV} of height ranged between 0.159-1.176 and that of BMI was between 0.157- 0.542³⁷.

Compared to the literature values, h^2_{CV} for height (0.713; 95% CI: 0.690,0.736) and BMI (0.318; 95% CI: 0.296,0.341) using RARity are on the higher spectrum of what has been reported in the literature, but

neither are they the highest estimates reported. The estimates observed may be due to the specific subset of the population selected in this study.

We highly appreciate the reviewer's insight and have made changes to the manuscript to reflect these views.

METHODS

Genotype data quality control

Common Variants:

Updated main text page 21; lines 474-478: We further filtered genotype data by LD pruning with $r^2 > 0.9$ and a rolling window of 1Mb, that were shifted in steps of 500 bases, leaving 1,030,594 CVs in 159,058 participants, for whom we also had WES data, which were used to construct blocks of 20,000 variants. These pruning parameters were selected based on simulations using a very similar approach, using 325,989 participants in the UKB by Di Scipio and Khan, et al¹⁶.

DISCUSSION

Updated main text page 13; lines 274-285: The performance RARity for CV contributions was comparable with BOLT. The estimated h^2_{CV} by LDSC on the other hand was lower than RARity for each trait. While LDSC is a powerful tool that allows estimation h^2_{CV} from GWAS summary-statistics, with adequate control for population stratification, it is prone to underestimation as discussed in several studies²⁸⁻³¹. Comparison of the h^2_{CV} for height and BMI, with other SNP-heritability methods in the literature, such as LDSC²³, LDAK³², GCTA³³, GRE¹⁹ and SumHer³⁴, shows that our estimates are on the higher end (but not the highest) of what has been historically reported in comparison studies^{19,30}. The apparent higher h^2_{CV} for height and BMI using RARity may be a characteristic of the subset of the population selected in this study. All current methods make assumptions about the genetic architecture, such as the effect size, variance, LD/MAF ratios, etc., since the true genetic architecture is unknown, it remains unclear which estimates in the literature are reliable. More importantly, none of these methods are suitable for estimation of rare variant heritability.

Comment 2.5. The current approach is only applicable to continuous traits. The authors mentioned that the extension to binary traits is future work, but it would be helpful if the authors provided a bit more information on how it can be done.

Response to Comment 2.5: *For dichotomous trait heritability, the model will largely be similar to that of the continuous trait but will require slight adjustments to transform the observed heritability to a liability scale heritability to account for the case-control imbalance in population-based studies⁴⁰. Thanks to the reviewer's suggestion, we have included this information in the manuscript.*

DISCUSSION

Updated main text page 17; lines 369-371: Currently, the RARity model is fitted for to continuous traits. For dichotomous trait heritability, the model will require slight adjustments to transform the observed heritability to a liability scale heritability to account for the case-control imbalance in population study³⁹.

Comment 2.6. It isn't clear whether the current approach can be applicable to WGS data. Some discussion would be helpful.

Response to Comment 2.6: *Thanks to the reviewer's suggestion, we now mention the possibility of using RARity for WGS in the manuscript (Please see **Response to Comment 2.1**). One of the challenges of dealing with WGS is that there is no natural biological unit to consolidate the intergenic regions. The most popular method is to use a sliding window, the consequences of which are also debated. With RARity, there is no need for defining the biological units, as the estimates can be performed on blocks that are agnostic of genetic borders, while maintaining the statistical model. However, we will need to calibrate RARity with further simulations to ensure accurate estimates, which will be worthwhile as larger sample sizes become accessible.*

Comment 2.7. The authors mentioned that one of the benefits of gene-based heritability estimation is that it can be used to find gene-level associations. But extensive literature exists for gene-based rare variant tests, and the method used in RARity (F-test) is less developed. For example, it cannot account for sample relatedness. In addition, some of the existing methods were developed to address the limitation of the F-test, which can have low power due to testing too many variants (large degree of freedom of the test). So I don't think RARity would be a useful tool for gene discovery as there are better alternatives.

Response to Comment 2.7: *We thank the reviewer for their insight. RARity is a method of estimating heritability, as such it is not meant to replace the current state-of-the-art association testing, but rather complement them, providing further insights that may be missed with association testing alone. In fact, to the best of our knowledge, this is the only method capable of estimating heritability for truly rare variants, from WES for large scale datasets. The F-test statistics was used to calculate p-values of the heritability estimates, and thereby identify genes with significant heritability. As demonstrated in the manuscript, being a versatile method that does not require parametrization nor assumptions regarding the genetic architecture of traits analyzed (such as polygenicity of effects or relationships between MAF/LD and effect size), makes it a powerful tool with many applications.*

References for Response to Reviewers' Comments:

1. Galinsky, K.J., Loh, P.R., Mallick, S., Patterson, N.J. & Price, A.L. Population Structure of UK Biobank and Ancient Eurasians Reveals Adaptation at Genes Influencing Blood Pressure. *Am J Hum Genet* **99**, 1130-1139 (2016).
2. Rawlik, K., Canela-Xandri, O. & Tenesa, A. Evidence for sex-specific genetic architectures across a spectrum of human complex traits. *Genome Biol* **17**, 166 (2016).
3. Gilks, W.P., Abbott, J.K. & Morrow, E.H. Sex differences in disease genetics: evidence, evolution, and detection. *Trends Genet* **30**, 453-63 (2014).
4. Traglia, M. *et al.* Genetic Mechanisms Leading to Sex Differences Across Common Diseases and Anthropometric Traits. *Genetics* **205**, 979-992 (2017).
5. Ge, T., Chen, C.Y., Neale, B.M., Sabuncu, M.R. & Smoller, J.W. Phenome-wide heritability analysis of the UK Biobank. *PLoS Genet* **13**, e1006711 (2017).
6. Bernabeu, E. *et al.* Sex differences in genetic architecture in the UK Biobank. *Nat Genet* **53**, 1283-1289 (2021).
7. Loh, P.R. *et al.* Efficient Bayesian mixed-model analysis increases association power in large cohorts. *Nat Genet* **47**, 284-90 (2015).
8. Bulik-Sullivan, B.K. *et al.* LD Score regression distinguishes confounding from polygenicity in genome-wide association studies. *Nat Genet* **47**, 291-5 (2015).
9. Park, L. Population-specific long-range linkage disequilibrium in the human genome and its influence on identifying common disease variants. *Sci Rep* **9**, 11380 (2019).
10. Taliun, D. *et al.* Sequencing of 53,831 diverse genomes from the NHLBI TOPMed Program. *Nature* **590**, 290-299 (2021).
11. Yang, J., Lee, S.H., Goddard, M.E. & Visscher, P.M. GCTA: a tool for genome-wide complex trait analysis. *Am J Hum Genet* **88**, 76-82 (2011).
12. Loh, P.R. *et al.* Contrasting genetic architectures of schizophrenia and other complex diseases using fast variance-components analysis. *Nat Genet* **47**, 1385-92 (2015).
13. Evans, L.M. *et al.* Comparison of methods that use whole genome data to estimate the heritability and genetic architecture of complex traits. *Nature Genetics* **50**, 737-745 (2018).
14. Ni, G., Moser, G., Wray, N.R., Lee, S.H. & Consortium, S.W.G.o.t.P.G. Estimation of Genetic Correlation via Linkage Disequilibrium Score Regression and Genomic Restricted Maximum Likelihood. *Am J Hum Genet* **102**, 1185-1194 (2018).
15. Srivastava, A.K., Williams, S.M. & Zhang, G. Heritability Estimation Approaches Utilizing Genome-Wide Data. *Curr Protoc* **3**, e734 (2023).
16. Di Scipio, M. *et al.* A versatile, fast and unbiased method for estimation of gene-by-environment interaction effects on biobank-scale datasets. *Nat Commun* **14**, 5196 (2023).
17. Wang, Q. *et al.* Landscape of multi-nucleotide variants in 125,748 human exomes and 15,708 genomes. *Nat Commun* **11**, 2539 (2020).
18. Coassin, S. & Kronenberg, F. Lipoprotein(a) beyond the kringle IV repeat polymorphism: The complexity of genetic variation in the LPA gene. *Atherosclerosis* **349**, 17-35 (2022).
19. Lee, S., Abecasis, R., Gonçalo, Boehnke, M. & Lin, X. Rare-Variant Association Analysis: Study Designs and Statistical Tests. *The American Journal of Human Genetics* **95**, 5-23 (2014).

20. Wu, M.C. *et al.* Rare-variant association testing for sequencing data with the sequence kernel association test. *Am J Hum Genet* **89**, 82-93 (2011).
21. Raudvere, U. *et al.* g:Profiler: a web server for functional enrichment analysis and conversions of gene lists (2019 update). *Nucleic Acids Res* **47**, W191-W198 (2019).
22. Shannon, P. *et al.* Cytoscape: a software environment for integrated models of biomolecular interaction networks. *Genome Res* **13**, 2498-504 (2003).
23. Piñero, J. *et al.* The DisGeNET knowledge platform for disease genomics: 2019 update. *Nucleic Acids Res* **48**, D845-D855 (2020).
24. Freshour, S.L. *et al.* Integration of the Drug-Gene Interaction Database (DGIdb 4.0) with open crowdsourcing efforts. *Nucleic Acids Res* **49**, D1144-D1151 (2021).
25. Hogue, J.C. *et al.* Differential effect of fenofibrate and atorvastatin on in vivo kinetics of apolipoproteins B-100 and B-48 in subjects with type 2 diabetes mellitus with marked hypertriglyceridemia. *Metabolism* **57**, 246-54 (2008).
26. Burch, K.S. *et al.* Partitioning gene-level contributions to complex-trait heritability by allele frequency identifies disease-relevant genes. *Am J Hum Genet* **109**, 692-709 (2022).
27. Zhang, Z., Xiao, X., Zhou, W., Zhu, D. & Amos, C.I. False positive findings during genome-wide association studies with imputation: influence of allele frequency and imputation accuracy. *Hum Mol Genet* **31**, 146-155 (2021).
28. Hernandez, R.D. *et al.* Ultrarare variants drive substantial cis heritability of human gene expression. *Nat Genet* **51**, 1349-1355 (2019).
29. Mancuso, N. *et al.* The contribution of rare variation to prostate cancer heritability. *Nat Genet* **48**, 30-5 (2016).
30. Marouli, E. *et al.* Rare and low-frequency coding variants alter human adult height. *Nature* **542**, 186-190 (2017).
31. Wainschein, P. *et al.* Assessing the contribution of rare variants to complex trait heritability from whole-genome sequence data. *Nat Genet* **54**, 263-273 (2022).
32. Yang, J. *et al.* Genetic variance estimation with imputed variants finds negligible missing heritability for human height and body mass index. *Nat Genet* **47**, 1114-20 (2015).
33. Jang, S.K. *et al.* Rare genetic variants explain missing heritability in smoking. *Nat Hum Behav* (2022).
34. Chen, W., Coombes, B.J. & Larson, N.B. Recent advances and challenges of rare variant association analysis in the biobank sequencing era. *Front Genet* **13**, 1014947 (2022).
35. Bocher, O. & Génin, E. Rare variant association testing in the non-coding genome. *Hum Genet* **139**, 1345-1362 (2020).
36. Hou, K. *et al.* Accurate estimation of SNP-heritability from biobank-scale data irrespective of genetic architecture. *Nat Genet* **51**, 1244-1251 (2019).
37. Evans, L.M. *et al.* Comparison of methods that use whole genome data to estimate the heritability and genetic architecture of complex traits. *Nat Genet* **50**, 737-745 (2018).
38. Speed, D. *et al.* Reevaluation of SNP heritability in complex human traits. *Nat Genet* **49**, 986-992 (2017).
39. Speed, D. & Balding, D.J. SumHer better estimates the SNP heritability of complex traits from summary statistics. *Nat Genet* **51**, 277-284 (2019).
40. Lee, S.H., Wray, N.R., Goddard, M.E. & Visscher, P.M. Estimating missing heritability for disease from genome-wide association studies. *Am J Hum Genet* **88**, 294-305 (2011).

REVIEWERS' COMMENTS

Reviewer #1 (Remarks to the Author):

The authors have done an excellent job to address all my comments and concerns. I have no further questions. The method would be a great addition to the field.

Reviewer #2 (Remarks to the Author):

The authors addressed all my comments and I don't have any additional ones.